# OSDA Agent: Leveraging Large Language Models for De Novo Design of Organic Structure Directing Agents

**Zhaolin Hu**[1,2] , **Yixiao Zhou**[2,3], **Zhongan Wang**[2], **Xin Li** [4*],**Weimin Yang**[4]
**Hehe Fan**[2], **Yi Yang**[1,2*]

[1]The State Key Laboratory of Brain-Machine Intelligence, Zhejiang University, China
[2]CCAI, Zhejiang University, China    [3]Shanghai Innovation Institute, China
[4]The State Key Laboratory of Green Chemical Engineering and Industrial Catalysis,
Sinopec Shanghai Research Institute of Petrochemical Technology, China
{12321165,12421181,zhonganwang,hehefan,yangyics}@zju.edu.cn
{lixin.sshy,yangwm.sshy}@sinopec.com

## Abstract

Zeolites are crystalline porous materials that have been widely utilized in petrochemical industries as well as sustainable chemistry areas. Synthesis of zeolites often requires small molecules termed Organic Structure Directing Agents (OSDAs), which are critical in forming the porous structure. Molecule generation models can aid the design of OSDAs, but they are limited by single functionality and lack of interactivity. Meanwhile, large language models (LLMs) such as GPT-4, as general-purpose artificial intelligence systems, excel in instruction comprehension, logical reasoning, and interactive communication. However, LLMs lack in-depth chemistry knowledge and first-principle computation capabilities, resulting in uncontrollable outcomes even after fine-tuning. In this paper, we propose OSDA Agent, an interactive OSDA design framework that leverages LLMs as the brain, coupled with computational chemistry tools. The OSDA Agent consists of three main components: the Actor, responsible for generating potential OSDA structures; the Evaluator, which assesses and scores the generated OSDAs using computational chemistry tools; and the Self-reflector, which produces reflective summaries based on the Evaluator's feedback to refine the Actor's subsequent outputs. Experiments on representative zeolite frameworks show the generation-evaluation-reflection-refinement workflow can perform de novo design of OSDAs with superior generation quality than the pure LLM model, generating candidates consistent with experimentally validated OSDAs and optimizing known OSDAs.

## 1 Introduction

Zeolites, a class of microporous silicate-based materials, have been widely used as highly efficient catalysts and adsorbents, in petrochemical industries and sustainable chemistry processes (Davis, 2002). The unique properties of zeolites stem from their porous frameworks and the synthesis of zeolites often requires the use of small molecules named Organic Structure Directing Agents (OS-DAs) (Moliner et al., 2013). OSDAs act as templates influencing the size, shape, and connectivity of the pores of zeolites. The design of effective OSDAs is crucial for tailoring the properties of the target zeolite, but this process has traditionally been guided by empirical knowledge and labor-intensive trial-and-error methods.

Recently, artificial intelligence has been extensively applied across various scientific disciplines, including chemistry (Liu et al., 2023; Jiang et al., 2024), biology (Fan et al., 2022; Corso et al.; Fan et al., 2023), and materials (Sriram et al., 2024; Gruver et al., 2024; Yang et al., 2024; Zeni et al., 2023). In particular, artificial intelligence has significantly influenced the design of OSDAs. Various heuristic algorithms, such as genetic algorithms (Pophale et al., 2013) and ant colony algo-

---

*Yi Yang and Xin Li are the corresponding authors.

rithms (Muraoka et al., 2020), have been employed to explore the vast chemical space of potential OSDAs, identifying promising candidates. Machine learning has also made significant strides in this area (Chong et al., 2020; Xie et al., 2019; Manuel Serra et al., 2007; Moliner et al., 2019). With the development of large-scale zeolite synthesis datasets (Pan et al., 2024; Muraoka et al., 2019; Jensen et al., 2021), neural networks trained on these extensive datasets can predict the suitability of OSDAs for specific zeolite structures. These networks effectively learn from established synthesis recipes and experimental results, enabling the design of OSDA molecules (Jensen et al., 2021; Xu et al., 2023). Although these methods enable OSDA predictions, their inability to accept feedback from human experts limits the interactability of the model.

Large language models (LLMs), such as GPT-4 developed by OpenAI (OpenAI, 2023), are capable of processing and generating human-like text and performing complex reasoning based on extensive datasets. These models have demonstrated remarkable proficiency in various natural language processing and multimodal tasks, including text generation, translation, summarization, and multi-modal understanding (Raffel et al., 2020; Brown, 2020; Devlin, 2018; Li et al.; Zhang et al., 2024; Li et al., 2025). OSDA molecular design based on LLM can provide a more natural and seamless user experience that goes beyond traditional machine learning methods, with more aspects of the model accepting suggestions from chemical experts, leading to better generation (Ito et al., 2024b;a). While LLMs can process vast amounts of data and generate novel molecular structures, they often lack the domain-specific reasoning and ability to effectively integrate complex experimental constraints or optimization criteria (Burtsev et al., 2023).

In this paper, we introduce the OSDA Agent, an innovative interactive framework for the design of OSDAs that harnesses the capabilities of large language models as the core intelligence, complemented by advanced computational chemistry tools. The OSDA Agent comprises three key components: the Actor, tasked with generating a diverse array of potential OSDA structures based on predefined criteria, utilizing the generative capabilities of LLMs to explore vast chemical space and synthesize novel molecular architectures; the Evaluator, which critically assesses and scores these OSDAs using a suite of computational chemistry tools (Landrum, 2013; Coley et al., 2018), employing various metrics to determine the feasibility and effectiveness of each proposed structure; and the Self-reflector, which plays a pivotal role in the iterative design process by producing comprehensive reflective summaries (Shinn et al., 2024) based on the Evaluator's feedback, guiding the Actor in refining its outputs to enhance the quality of subsequent designs. Through the integration of these components, the OSDA Agent not only streamlines the design process but also fosters a continuous learning environment, enabling the system to adapt and improve over time, representing a significant advancement in the automated design of OSDAs and facilitating a more efficient and effective approach to zeolite synthesis. In summary, our contributions are threefold:

- We introduce the OSDA Agent, an innovative interactive framework that integrates LLMs with computational chemistry tools for the de novo design of OSDAs given target zeolites.

- To cope with the problem of uncontrollable results generated by LLMs, we introduced a reflection mechanism that consists of a novel binding-energy prediction module and chemical traditional tools.

- Our experimental results demonstrate that the OSDA Agent significantly improves the quality of OSDA candidates compared to traditional methods, yielding structures that align with experimentally validated OSDAs and effectively optimizing known OSDAs.

## 2 RELATED WORK

**Designing OSDAs for Zeolites.** The interaction between OSDAs and zeolites can be captured through atomistic simulations, such as density functional theory (DFT) and molecular dynamics simulations, though these methods require substantial computational resources (Schwalbe-Koda & Gómez-Bombarelli, 2021a). Jensen et al. (Jensen et al., 2021) utilized data-driven approaches to explore relationships between OSDAs, qualitative gel chemistry, and zeolite structures. Daeyaert et al. (Daeyaert et al., 2019) and Xu et al. (Xu et al., 2023) applied machine learning to train neural networks for designing OSDAs tailored to specific zeolite structures. Recent studies by Ito et al. (Ito et al., 2024b;a) investigated the use of LLMs in designing novel OSDA molecules, highlighting their potential for advancing molecular innovation. In contrast to previous approaches, our method combines the knowledge embedded in LLMs with chemical tools to improve design outcomes.

**LLM Agents for Science.** Large language model (LLM) agents are now widely used across various scientific disciplines (Chiang et al., 2024; Huang et al., 2024; Skarlinski et al., 2024; Ma et al., 2024; Ghafarollahi & Buehler, 2024; Lála et al., 2023; Boiko et al., 2023; Darvish et al., 2024). ChemCrow (M. Bran et al., 2024) developed an LLM agent integrated with specialized tools for literature retrieval, molecular modification, and reaction execution, enabling the autonomous execution of chemical synthesis. Several studies (Buehler, 2024; Sprueill et al., 2024; Ansari & Moosavi, 2023; Kang & Kim, 2024) have explored the application of LLM agents in the field of materials science. Furthermore, MedAgents (Tang et al., 2023) and DRUGAGENT (Inoue et al., 2024) have explored the applications of LLM Agents in the fields of medicine and drug discovery. In this work, we investigate the application of an LLM Agent for OSDA molecule design.

**Text-based de novo Molecule Generation.** In recent decades, AI-driven methods have become central to advancing molecule discovery (Hu et al., 2023; Fan & Yang, 2024). Text-based de novo molecule generation utilizes natural language processing techniques alongside chemical information to design novel molecular structures. Text2Mol (Edwards et al., 2021), constructs paired datasets of molecules and textual descriptions to learn a shared semantic embedding space for improved retrieval. MolT5 (Edwards et al., 2022) and BioT5 (Pei et al., 2023), built on the T5 (Raffel et al., 2020) architecture, are pre-trained on large corpora of text and molecular strings, enhancing their ability to generate molecules from textual input. MoMu (Su et al., 2022) uses contrastive learning to bridge molecular graphs and related text data. TGM-DLM (Gong et al., 2024) explores diffusion models for molecular generation. Moreover, FrontierX (Sakhinana & Runkana, 2023) and Mol-ReGPT (Li et al., 2024) utilize the capabilities of large language models, presenting new paradigms for molecular generation. However, traditional text-to-molecule methods struggle to perform well when faced with the complexity of designing OSDA molecules. In this work, we address this challenge by utilizing the comprehensive capabilities of the LLM Agent.

## 3 PRELIMINARIES AND DATA DESCRIPTION

### 3.1 DATA

We utilized the Zeolite Organic Structure Directing Agent Database (OSDB) (Schwalbe-Koda et al., 2021) that includes 112,400 OSDA-Zeolite pairs which comprises 549 OSDA molecules and 209 distinct zeolite frameworks extracted from the literature. The database provides the corresponding binding energy for each OSDA-Zeolite pair along with the OSDA-Zeolite complex structure data saved in CIF files. In addition, for each OSDA molecule, we calculated its synthetic complexity score (SCScore) (Coley et al., 2018).

Furthermore, we employed the Jensen dataset (Jensen et al., 2021) extracted from the entire body of zeolite literature, spanning over 140 journals from more than 15 publishers between 1966 and 2020. It includes 5,663 synthesis routes extracted from 1,384 articles, encompassing OSDA molecules, synthesis gel compositions, and the resulting zeolite phases. The dataset provides details on 758 distinct OSDA molecules and 205 zeolite phases, from which we extracted OSDA molecules as real-world examples of existing OSDAs.

### 3.2 THE STRUCTURE OF OSDA-ZEOLITE COMPLEX

The OSDA-Zeolite complex is a variation of crystal structure that still adheres to the geometric constraints of crystals. Thus, we represent it as a crystal graph. In the unit cell $U = (X, P)$, $X = [x_1, x_2, \cdots, x_{n-1}, x_n]^T \in \mathbb{R}^{n \times 1}$, where $n$ represents the number of atoms, and $x_i \in \mathbb{R}^1$ represents the type of atom $i$ in the unit cell. $P = [p_1, p_2, \cdots, p_{n-1}, p_n]^T \in \mathbb{R}^{n \times 3}$ is the atomic position matrix, where $p_i \in \mathbb{R}^3$ represents the Cartesian coordinates of atom $i$ in the unit cell in three-dimensional space. To further encode the periodic pattern, additional lattice vectors $L = [l_1, l_2, l_3]^T \in \mathbb{R}^{3 \times 3}$ are used to describe how the unit cell repeats in three directions. Thus, in 3D space, we represent the OSDA-Zeolite complex as $(X, P, L)$, see Figure 11 in Appendix E.

### 3.3 BINDING ENERGY

The affinity of an OSDA-Zeolite pair can be quantitatively reflected via the binding energy. A smaller binding energy value indicates that the OSDA is more compatible with the zeolite. The

calculation formula of binding energy is:

$$E_b = E_{\text{zeo-osda}} - E_{\text{zeo}} - E_{\text{osda}} \tag{1}$$

where $E_b$ represents the binding energy, $E_{\text{zeo-osda}}$ is the energy of the OSDA-Zeolite complex, $E_{\text{zeo}}$ is the energy of the zeolite, and $E_{\text{osda}}$ is the energy of the OSDA (Schwalbe-Koda & Gómez-Bombarelli, 2021a).

## 4 METHODOLOGIES

### 4.1 OSDA AGENT FRAMEWORK

In developing the OSDA Agent framework, we followed the methodology described in (Shinn et al., 2024). Our framework integrates three interconnected models: the Actor, the Evaluator, and the Self-reflection model. The Actor is responsible for creating OSDAs by employing advanced algorithms that guide molecules into desired configurations. The Evaluator assesses the OSDA molecules generated by the Actor, utilizing various chemical tools for evaluation. The Self-reflection model leverages feedback from the Evaluator to refine the synthesis processes within the Actor, continuously improving the efficacy of the organic structure-directing agents. This cohesive approach provides a robust framework for generating and optimizing OSDAs. The overall framework architecture is illustrated in Figure 1 (a).

More specifically, the Actor in this work assumes the role of a zeolite expert. The Actor is based on a LLM that generates SMILES sequences for OSDA molecules in response to specified prompts and requirements. In this context, we utilize In-Context Learning (Yoo et al., 2021) and Chain of Thought (Wei et al., 2022) techniques to guide the Actor's performance. The specific prompt strategies we employ are detailed in Section 4.2. Additionally, we incorporate a memory component, which provides supplementary context to enhance the Actor's capabilities.

The Evaluator component of the OSDA Agent framework plays a critical role in assessing the quality of the OSDA molecules generated by the Actor. It takes the candidate SMILES sequences produced by the Actor as input and employs a range of chemical tools for evaluation. The chemical tools we use consist of a series of small models and predefined rules. First, RDKit is utilized to verify that the generated SMILES expressions adhere to chemical validity. Then, the Evaluator provides a comprehensive assessment by scoring the potential OSDA molecules based on their binding energy and SCScore (Coley et al., 2018). This approach ensures that the generated OSDA molecules meet the required specifications and are synthetically feasible, as detailed in Section 4.3.

Although LLMs exhibit powerful intelligence, they occasionally make reasoning errors (Madaan et al., 2024). To mitigate such errors, we implemented a self-reflection module. Based on a LLM, this module provides valuable feedback for future iterations through verbal self-reflection. It generates detailed and specific feedback based on the assessments from the Evaluator, the current generated results, and the memory of the agent. In subsequent trials, the agent leverages its past experiences to optimize the generation of OSDA molecules. This iterative process of trial and error, self-reflection, and persistent memory enables the agent to rapidly enhance its decision-making abilities across various environments by effectively utilizing feedback signals.

### 4.2 STRUCTURE DESIGN WITH COT STRATAGY

To address the limitations of LLMs in handling complex reasoning tasks, we adopted a few-shot Chain of Thought (few-shot CoT) approach. By designing specialized prompt strategies, we aim to foster a more continuous and incremental reasoning process within the model, enabling it to more effectively generate valid OSDA molecules.

The few-shot CoT prompt combines In-Context Learning (ICL) (Brown, 2020) with the Chain of Thought (CoT) (Wei et al., 2022) technique. By introducing a continuous, step-by-step reasoning process through the prompt "Let's think step by step," we effectively guide the model in solving complex tasks. CoT enables the model to break down tasks into their constituent parts, leading to clearer and more logical solutions (Wei et al., 2022). As a few-shot prompt, we also provide the LLM with an example to help it better understand and respond to the required format and content.

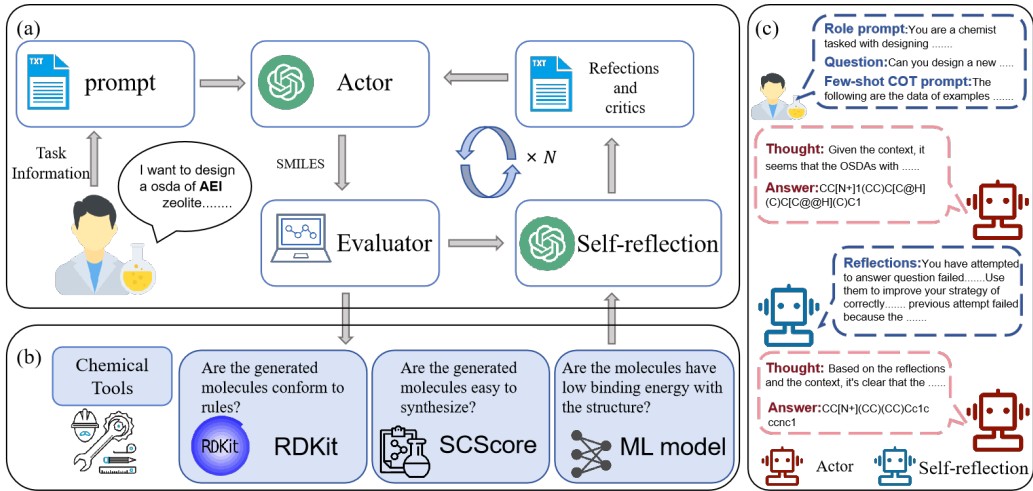

Figure 1: OSDA Agent Framework. (a) The OSDA Agent pipeline, integrating three models: Actor, Evaluator, and Self-reflector, responsible for generating molecules, performing chemical evaluations, and providing optimization feedback. (b) The Evaluator uses three chemical tools to assess molecules based on chemical principles, synthetic feasibility, and binding energy. (c) Example of OSDA Agent in action.

In crafting the prompts for the generative large language model, we also employed role-play instructions, assigning the model the role of an expert in zeolite OSDAs. Additionally, we extracted several well-performing OSDA molecules from the OSDB database to serve as context. The complete prompt can be found in Figure 7 and Figure 8 in Appendix D.

## 4.3 EVALUATION WITH CHEMICAL TOOLS

The LLM Agent leverages its powerful natural language processing capabilities to intelligently generate molecular structures. However, due to the intricate chemical constraints and reasoning requirements, the output generated by the LLM alone may not fully meet the precision required in chemistry. Therefore, we introduced a series of chemical tools to ensure that the generated molecules adhere to chemical rules and possess synthetic feasibility. This combination provides us with a robust and flexible framework for exploring and optimizing new OSDA molecules.

We incorporated three key chemical tools to help the agent improve the chemical validity and accuracy of the generated OSDA molecules. These tools include the RDKit (Landrum, 2013) toolkit, SCScore (Coley et al., 2018), and the binding energy estimation model. In the following sections, we will provide a detailed introduction to each of these tools.

We also developed a memory component that stores the evaluation results from the chemical tools for the generated molecules. These stored evaluations are utilized by the Reflection Mechanism to inform and improve subsequent iterations.

### 4.3.1 RDKIT TOOLKIT

The LLMs occasionally generate OSDA molecules that do not meet the required standards, including violations of chemical rules and OSDA-specific empirical guidelines. To address this issue, we used the RDKit (Landrum, 2013) toolkit to evaluate and filter the generated molecules. The specific empirical criteria, similar to those outlined in (Ito et al., 2024b), are listed in Table 2 in Appendix C.

### 4.3.2 SYNTHETIC COMPLEXITY SCORE

The synthetic accessibility of OSDAs is a crucial factor in assessing their quality. In our work, we employed SCScore to assess synthetic complexity. SCScore is a scoring method based on reaction database learning, where a neural network model is trained on a large dataset of 12 million reactions from the Reaxys database to predict the number of reaction steps required to synthesize the target

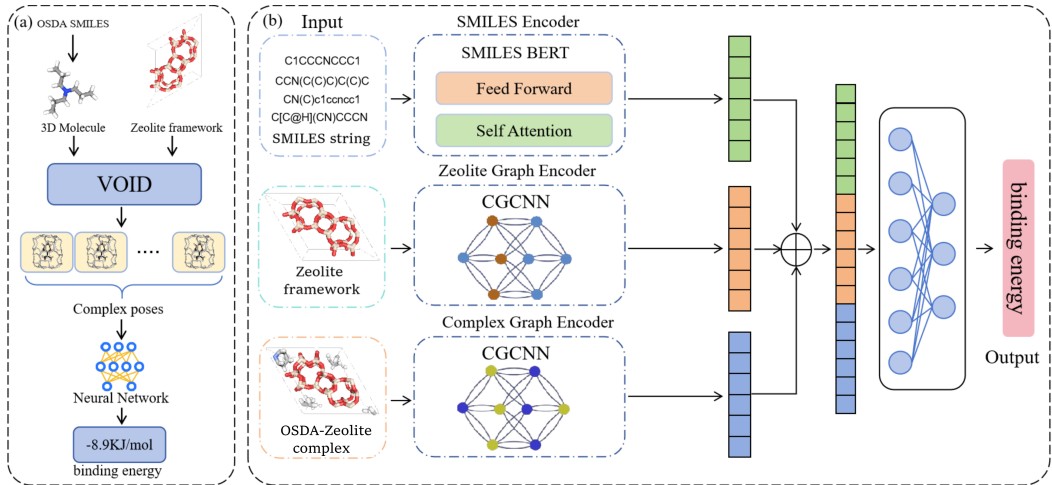

Figure 2: Details of the binding energy estimation model. (a) Process: We provide ⟨OSDA, Zeolite⟩ pairs to the VOID tool to generate docking poses. The predicted energy for each pose is then computed using a neural network model, and the lowest is selected as the estimated binding energy. (b) Neural network architecture: The input consists of three types of information relevant to determining the binding energy: OSDA, Zeolite, and Complex. We use different architectures to extract information from each and then integrate them to predict the binding energy.

molecule (Coley et al., 2018; Lawson et al., 2014). The model establishes a nonlinear metric by comparing the synthetic complexities of reactants and products, effectively capturing the increasing complexity across multi-step synthesis routes. The SCScore ranges from 1 to 5, with lower values indicating that a molecule is easier to synthesize.

### 4.3.3 BINDING ENERGY ESTIMATION MODEL

Since each OSDA Agent design involves evaluating the quality of the OSDA design multiple times using binding energy, traditional atomic simulation methods for calculating binding energy often require computing the energy of dozens to hundreds of complexes, which is time-consuming (Schwalbe-Koda & Gómez-Bombarelli, 2021a). To address this issue, we proposed a deep learning-based method to estimate the energy of these complexes, significantly reducing the time required for binding energy calculations, as shown in Figure 2 (b).

Our inspiration stems from the calculation formula for the OSDA-Zeolite binding energy. In Equation equation 1, the binding energy is related to the energy of complex, the energy of zeolite, and the energy of OSDA. We follow this principle in our approach. The inputs to our method include molecular SMILES, zeolite structure diagrams, and complex structure diagrams. The zeolite structure diagrams are obtained by parsing CIF files. We obtain the complex's CIF files using the VOID (Voronoi Organic-Inorganic Docker package) (Schwalbe-Koda & Gómez-Bombarelli, 2021b) and parse them to extract the complex structure diagrams.

We utilize a pre-trained SMILES-BERT (Wang et al., 2019) model to extract molecular features, while the complex and zeolite structure diagrams are processed using CGCNN (Xie & Grossman, 2018) networks to extract complex and structural features. Then, these features (molecular, structural, and complex) are concatenated, followed by feature fusion using a transformer, and finally, a fully connected neural network is used to predict the complex energy. Our model is trained on data from the OSDB database. The effectiveness of each module is shown in Table 3 in Appendix E.

Throughout the entire process, candidate OSDA molecular SMILES are generated using Actor, and the 3D conformations of the molecules are optimized using the MMFF94 (Halgren, 1996) method. Possible binding poses between the 3D molecular conformations and zeolite structures are generated using the VOID tool. We predict the energy for each pose and use the lowest energy as the estimated binding energy, as illustrated in Figure 2 (a).

## 4.4 REFLECTION MECHANISMS

The Reflection Mechanism reads the evaluations of the OSDA molecules generated by the Actor, stored in the memory component. Using a specially designed reflection prompt, the Self-reflection LLM provides constructive feedback on the Actor's output. This process repeats for a set number of iterations, guiding the Actor to improve the quality and success rate of the generated molecules.

The reflection prompt utilizes a few-shot prompting technique, providing the Self-reflection LLM with several examples of constructive feedback. We instruct the Self-reflection LLM to critique and offer suggestions on the generated results based on the evaluations from the chemical tools. The feedback can focus on several aspects, including the chemical validity of the generated molecules, their synthetic feasibility, and the estimated binding energy with the given zeolite structure. The goal is to produce OSDA molecules that are both effective and easy to synthesize. The complete prompt and examples can be found in Figure 9 and Figure 10 in Appendix D.

## 5 EXPERIMENTS

### 5.1 EXPERIMENT SETUP

In our OSDA Agent framework, the Actor utilizes the GPT-4 model, while the Self-reflection is based on the GPT-4o model. The examples used in the few-shot CoT prompts are sourced from the OSDB database. We built our framework using Langchain, and the RDKit version employed is 2023.9.1. In the experiments, the literature OSDAs were obtained from the Jensen dataset (Jensen et al., 2021). Baseline methods include state-of-the-art text-based de novo molecule generation approaches, such as BioT5 (Pei et al., 2023), MolT5 (Edwards et al., 2022).

The binding energy estimation model was trained on OSDB data. The model was trained for 15 epochs using the Adam optimizer, with a 1e-5 learning rate and batch size of 32. It was implemented in Python v3.10.4 with the PyTorch v1.12 framework and trained on an NVIDIA A6000 GPU. To ensure the independence of the training and test sets, the dataset was split based on SMILES, following an 8:1:1 ratio. This split ensured that the OSDA molecules in the training set were completely distinct from those in the test set. The details and results can be found in Appendix E.

### 5.2 RESULTS OF OSDA MOLECULE DESIGN

To validate the effectiveness of our approach, we selected two representative zeolite frameworks for testing: the small-pore LTA framework (Reed & Breck, 1956) with a cage-like structure and the large-pore one-dimensional AFI framework (Bennett et al., 1983). We applied our method to design OSDA molecules tailored for these two zeolite structures. In this section, we focus on presenting the results and analysis of our method in designing OSDAs for the cage-like small-pore LTA framework and the large-pore one-dimensional AFI framework. Additionally, we extended our method to test other zeolite frameworks, including LEV (Barrer & Kerr, 1959), AEI (Simmen et al., 1991), AFX (McGuire et al., 1995), ITE (Camblor et al., 1997) and MOR (Simoncic & Armbruster, 2004), with the results provided in Appendix I.

#### 5.2.1 SUGGESTED OSDA CANDIDATES

Figure 3 presents several OSDAs predicted by our OSDA Agent using a de novo molecular design workflow. Compared to previous work (Ito et al., 2024b;a), our method incorporates synthetic feasibility into the model, resulting in OSDA candidates that not only match known OSDAs but also exhibit a high degree of synthesizability. Additionally, consultations with experts in zeolite chemistry affirmed that the generated molecules exhibit significant potential to function as OSDAs.

#### 5.2.2 DIMENSIONALITY REDUCTION ANALYSIS

An essential capability for any molecular generation model is to capture the molecular distribution and generate diverse and realistic molecules. Such capabilities are paramount when constructing virtual libraries to advance computer-aided drug discovery endeavors (van Hilten et al., 2019). To evaluate the performance of our model in generating OSDAs, we performed a comparative analysis

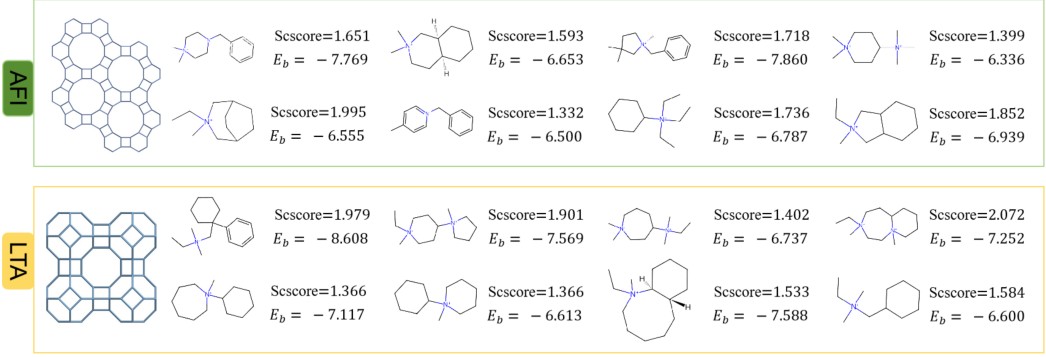

Figure 3: Examples of OSDAs for two representative zeolite frameworks, large-pore AFI framework and small-pore LTA framework, generated using the OSDA Agent. The $E_b$ is measured in kJ/mol Si.

against previously reported OSDAs using Principal Component Analysis (PCA) based on Weighted Holistic Invariant Molecular (WHIM) (Todeschini & Gramatica, 1997) descriptors (detailed in Appendix C).

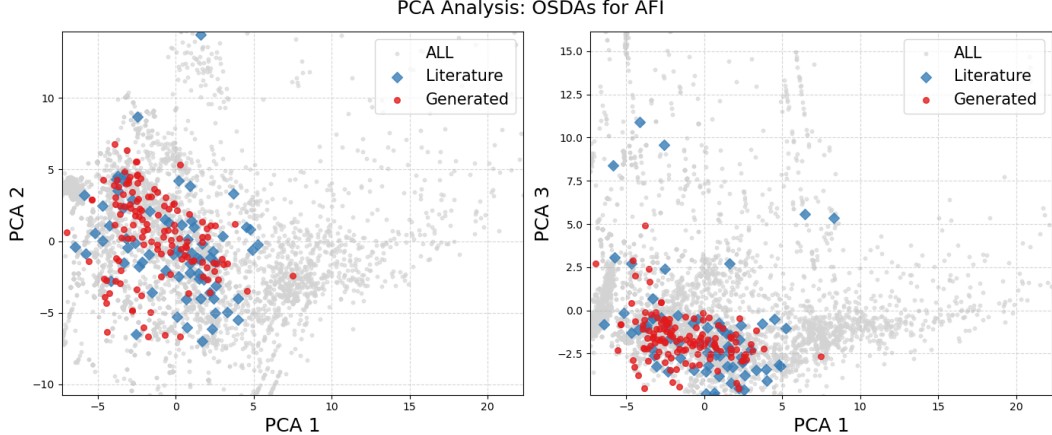

Figure 4: Principal Component Analysis (PCA) of WHIM vector representation for OSDA molecules used in AFI framework. PCA 1, 2, and 3 represent the first three principal axes. Red points show OSDAs generated by the OSDA Agent, blue points represent OSDAs of AFI framework from the literature, and gray points represent all OSDAs extracted from the literature.

The dataset used for this analysis includes all OSDAs from the literature (shown in gray), OSDAs generated by our model (shown in red), and OSDAs of specific frameworks of zeolite previously reported in the Jensen dataset (shown in blue). WHIM descriptors, which capture molecular geometry based on principal components derived from molecular field methods, were calculated for both datasets. These descriptors were then standardized and processed via PCA to visualize and assess the overlap between the OSDAs generated by our model and those previously reported in the reduced-dimensionality space.

Consistent with previous work(Jensen et al., 2021; Xu et al., 2023), we used PCA to reduce the WHIM descriptors to three dimensions. The explained variance of the principal components confirms that PCA 1, PCA 2, and PCA 3 capture the majority of the variance in the data (Figure 6 in Appendix C). We conducted tests on AFI and LTA frameworks, with the PCA results shown in Figures 4 and Figure 14 in Appendix F, displaying data points projected onto the first three principal components. The left plot (PCA 1 vs. PCA 2) illustrates the distribution of all OSDAs from the Jensen dataset (in gray), OSDAs generated by our model (in red), and OSDAs of AFI and LTA

frameworks previously reported in the literature (in blue). The right plot (PCA 1 vs. PCA 3) offers a complementary view of the data distribution.

This visualization demonstrates a notable overlap between the generated OSDAs and the reported OSDAs, which indicates that our model has successfully generated molecules that closely align with known, experimentally validated OSDAs. Furthermore, the red points suggest that our model is also capable of generating novel OSDAs that explore new regions of the molecular descriptor space, highlighting the generative model's potential for de novo OSDA design.

### 5.2.3 PERFORMANCE COMPARISON

To evaluate the effectiveness of our OSDA Agent framework, we compared the results of our approach with two text-based de novo molecule generation methods, MolT5 and BioT5, as well as the results generated by the pure GPT-4 model without the additional enhancements provided by our integrated method. To ensure a fair comparison, we supplied the GPT-4 model with the same OSDB data used for prompts in the OSDA Agent. We generated approximately 100 OSDA molecules each for both AFI and LTA frameworks using the OSDA Agent and GPT-4, while MolT5 and BioT5 produced around 500 OSDA molecules each for both AFI and LTA frameworks.

In this study, we employed seven well-established metrics (detailed in Appendix C) to assess the capability of our approach to generate OSDA molecules that align with real-world distributions and empirical rules. The first metric is Validity, which evaluates whether the generated molecules comply with the empirical rules for OSDAs, as shown in Table 2 of Appendix C. Next, we utilized four commonly used text-to-molecule metrics BLEU (Papineni et al., 2002), Morgan, MACCS, and RDK to measure the similarity between the generated OSDA molecules and those reported in the literature. Finally, we applied the Energy Distance (ED) (Székely & Rizzo, 2013) and Kullback-Leibler (KL) divergence (Kullback & Leibler, 1951) to assess the distributional similarity between the generated OSDAs and the ones reported in the literature.

As shown in Table 1, traditional text-to-molecule methods like MolT5 (Edwards et al., 2022) and BioT5 (Pei et al., 2023) struggle to generate molecules that comply with OSDA rules, largely due to their lack of specialized knowledge in zeolite OSDAs. The molecules they generate significantly differ from those reported in the literature. While large language models such as GPT-4 possess the relevant knowledge to produce more plausible OSDA molecules, they lack the guidance of specialized chemical tools. As a result, GPT-4 generated OSDA molecules exhibit significant shortcomings in terms of accuracy and diversity. Specifically, more than 50% of the OSDAs generated by the pure GPT-4 model do not meet OSDA requirements. Although certain similarity metrics (e.g., Morgan) show relatively high scores, the generated molecules exhibit significant differences in their overall distribution compared to the OSDAs reported in the literature. This issue is clearly illustrated in Figures 15 in Appendix F, where the distributions of molecular volume, molecular weight, and asphericity are visualized. Notably, the molecular volume and weight of the purely GPT-4 generated molecules deviate substantially from those of the OSDAs found in the literature.

Our OSDA Agent integrates the GPT-4 model with targeted prompt strategies, chemical validation tools, and a reflection mechanism, consistently producing OSDAs with higher chemical relevance and synthesizability. The inclusion of the SCScore model and RDKit for chemical rule validation allows our framework to filter out unfeasible molecules early in the process. Additionally, the reflection mechanism iteratively refines the generated OSDAs, enhancing their alignment with known experimental data and synthesis routes. This capability enables the OSDA Agent to accurately generate molecules that conform to OSDA rules and exhibit greater similarity to those found in the literature. As shown in Table 1, our OSDA Agent outperforms other methods across the majority of similarity metrics. In terms of distribution, the OSDA molecules generated by the OSDA Agent exhibit lower KL divergence and Energy distance (ED) values. Therefore, we conclude that the OSDAs generated by the OSDA Agent are more representative of those found in real-world scenarios.

### 5.3 RESULTS OF OPTIMIZING EXISTING OSDA MOLECULES

In the previous sections, we have demonstrated the capability of our OSDA Agent in molecular design, and we hope that our approach can also be effective in optimizing existing OSDA molecules. To demonstrate the capability of our approach in refining complex and challenging synthesized molecules, we applied the model to several OSDAs previously reported in the literature. These

Table 1: The performance of the OSDA Agent in the de novo design of OSDAs.The **best** scores are in bold, and the second-best scores are underlined.

| Type | Method | Validity↑ | BLEU↑ | Morgan↑ | MACCS↑ | RDK↑ | ED↓ | KL Divergence↓ |
|------|--------|-----------|-------|---------|--------|------|-----|----------------|
| AFI | MolT5   (Edwards et al., 2022) | 0.000 | 0.204 | 0.157 | 0.556 | 0.384 | 266.4 | 7.753 |
|     | BioT5   (Pei et al., 2023) | 0.008 | 0.193 | 0.147 | 0.384 | 0.162 | 120.7 | 3.128 |
|     | GPT-4 | 0.440 | 0.554 | **0.332** | 0.750 | 0.591 | 4.061 | 0.653 |
|     | OSDA Agent | **1.000** | **0.577** | 0.308 | **0.781** | **0.608** | **2.567** | **0.588** |
| LTA | MolT5   (Edwards et al., 2022) | 0.000 | 0.266 | 0.151 | 0.520 | 0.297 | 189.4 | 2.985 |
|     | BioT5   (Pei et al., 2023) | 0.005 | 0.228 | 0.130 | 0.431 | 0.337 | 73.43 | 1.093 |
|     | GPT-4 | 0.471 | 0.382 | 0.213 | 0.677 | 0.360 | 12.80 | 0.976 |
|     | OSDA Agent | **1.000** | **0.499** | **0.229** | **0.737** | **0.412** | **2.084** | 0.744 |

molecules, selected from the OSDB database, were chosen specifically for their complexity and synthesis difficulty. The goal was to enhance synthetic feasibility while maintaining low binding energies.

We tested the ability of our OSDA Agent to optimize existing OSDA molecules for AFI framework. We selected ten OSDA molecules from the OSDB database with relatively high SCScore, averaging around 3.45. Using our approach, we modified the prompt to "Can you start with this molecule and optimize this OSDA molecule...?" and set the reflection steps to 4 for each molecule. To ensure the reliability of the results, we repeated the optimization process five times for each of the molecules.

We calculated the average SCScore and binding energy across the five experiments, as shown in Figure 5. The results indicate that the average SCScore decreased from 3.45 to 2.46, suggesting a lower synthesis difficulty. The optimized binding energy averaged -6.71 kcal/mol, which remains within the reasonable range (-3.38 kcal/mol to -9.00 kcal/mol)

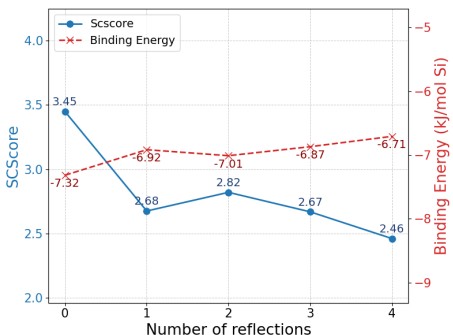

Figure 5: During the molecular optimization process, the SCScore and Binding Energy of the OSDA molecules varied with the number of reflection iterations.

based on the Jensen dataset. Additionally, the optimized molecules maintained structural similarity to the originals, preserving the functional properties necessary for zeolite templating. Detailed optimization results can be found in Appendix F, Figure 17.

The results indicate that our OSDA Agent successfully generated optimized versions of these molecules, reducing the SCScore and thereby lowering the synthesis difficulty, while preserving the functional characteristics necessary for zeolite templating. This demonstrates that the OSDA Agent has the potential not only to design new molecules but also to improve existing ones.

## 6    CONCLUSION

In this work, we introduced the OSDA Agent, which effectively designs and optimizes OSDAs through iterative refinement with targeted prompts. Applied to small and large pore frameworks, including LTA and AFI, the OSDA Agent generated chemically valid and structurally compatible molecules. Comparisons with literature-reported OSDAs revealed competitive, and in some cases superior, designs. These results highlight the OSDA Agent's potential for molecular design and optimization. Future work will focus on expanding the applicability of our methods to a broader range of scientific fields and further integrating experimental constraints into the optimization process.

## ACKNOWLEDGMENTS

This work are supported by National Science and Technology Major Project (2023ZD0120801), National Natural Science Foundation of China (62472381) and Fundamental Research Funds for the Zhejiang Provincial Universities (226-2024-00208).

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

## A  LINK TO DATASETS AND MODELS

Jensen dataset: `https://github.com/olivettigroup/OSDA_Generator`

OSDB database: `https://zeodb.mit.edu/index`

Pre-trained SMILES-BERT: `https://huggingface.co/unikei/bert-base-smiles`

GPT-4: `https://openai.com/chatgpt/`

VOID: `https://github.com/learningmatter-mit/VOID`

Mistral-Nemo-Instruct-2407:`https://huggingface.co/mistralai/Mistral-Nemo-Instruct-2407`

Llama-3.1-70B-Instruct:`https://huggingface.co/meta-llama/Llama-3.1-70B-Instruct`

## B  SUPPLEMENTING THE BACKGROUND AND MOTIVATION

**The Importance of OSDA in Zeolite Synthesis**
In zeolite synthesis, Organic Structure-Directing Agents (OSDAs) are essential in determining the final framework structure and morphology of the material. Zeolites are microporous materials with a wide range of applications in catalysis, adsorption, and ion exchange. The specific properties of zeolites, such as pore size and shape, depend on their crystalline structure, which is directly influenced by the presence of OSDAs. These organic molecules act as templates during the synthesis process, guiding the formation of the zeolite framework and ensuring the desired structural arrangement.

OSDAs not only direct the crystallization of zeolite but also contribute to the stability of the resulting material. In some cases, OSDAs function as charge-balancing agents, helping to maintain the integrity of the aluminosilicate framework during synthesis. Without the appropriate OSDA, zeolite formation may either fail to occur or result in amorphous materials with poorly defined structures. Therefore, the choice of OSDA is critical in controlling the final zeolite structure, affecting both the framework topology and the material's performance.

**Why do the design of OSDAs currently present significant challenges?**
OSDA plays a crucial role in determining the topology of zeolites, especially in the formation of their porous structures. Traditional methods mainly rely on two approaches: first, using empirical experience and experimentation to search for feasible OSDA molecules within a molecular library; and second, employing computational simulations of zeolite-OSDA interactions (such as Density Functional Theory, DFT) to assist in OSDA design. Both experience-based and simulation-based traditional methods are highly time-consuming and resource-intensive. In particular, the vast chemical space of OSDA (small molecules) presents significant challenges in discovering new OSDA candidates using conventional methods.

Moreover, due to the difficulty in finding new OSDA molecules, the existing pool of OSDA candidates is very limited. For example, the Jensen dataset, which collects data from papers published between 1966 and 2020, contains only 758 different OSDA molecules. Traditional machine learning generation methods typically require large datasets, but the small number of OSDA molecules makes it challenging to train effective generative models. Several traditional generative model approaches(Jensen et al., 2021; Xu et al., 2023) are typically trained using synthetic routes. This means that when designing OSDAs, information related to the synthetic pathway, such as gel chemistry, must be provided. Moreover, the limited number of different OSDA molecules in the dataset negatively impacts the model's ability to search effectively within the chemical space. Currently, Large Language Models (LLMs) have broad knowledge across related fields, and we aim to leverage the domain expertise of LLMs to overcome the data scarcity issue.

**Why is an iterative, feedback-based approach employed in this context?**
Due to the phenomenon of "hallucination" in large language models (LLMs), especially when performing complex tasks like molecular design, it is necessary to introduce additional chemical knowledge to help the LLMs accomplish the task more effectively. Through an interactive, feedback-driven approach, we professionally evaluate the molecules designed by the LLM, including factors

such as OSDA empirical rules, molecular synthesis difficulty, safety, etc., and provide feedback to the LLM. This process is similar to how chemists validate through experiments and make improvements based on the experimental results.

## C EXPERIMENTAL DETAILS AND METRICS

We outline the metrics employed to evaluate the performance of the generative models in our experiments, encompassing:

**Validity.** The proportion of generated molecules that follow the empirical rules for OSDAs can be expressed as:

$$V = \frac{N_{\text{valid}}}{N_{\text{total}}},$$

where $N_{\text{valid}}$ is the number of valid molecules and $N_{\text{total}}$ is the total number of generated molecules (see Table 2).

**BLEU.** To assess the difference between the generated molecules and OSDAs reported in the literature, the evaluation is conducted by calculating the average BLEU score $B$ between the SMILES representations of the generated molecules and those of the closest OSDAs documented in the literature:

$$B = \frac{1}{N} \sum_{i=1}^{N} \text{BLEU}(S_{\text{gen},i}, S_{\text{OSDA},i^*}),$$

where $N$ is the number of generated molecules, $S_{\text{gen},i}$ is the SMILES representation of the $i$-th generated molecule, and $S_{\text{OSDA},i^*} = S_{\text{OSDA},j^*}$ with

$$j^* = \arg\max_{j} \text{BLEU}(S_{\text{gen},i}, S_{\text{OSDA},j}),$$

where $j$ iterates over all OSDAs, ensuring $S_{\text{OSDA},i^*}$ is the SMILES representation of the OSDA most similar to the $i$-th generated molecule.

**Morgan.** Morgan fingerprints, which generate feature vectors by calculating the local atomic environments in molecules, are used here to measure the difference between the generated molecules and the OSDAs reported in the literature. Specifically, this is done by calculating the similarity $S_{Morgan}$ between each generated molecule and its closest OSDA documented in the literature, followed by averaging these scores:

$$S_{Morgan} = \frac{1}{N} \sum_{i=1}^{N} \text{sim}(F_{\text{gen},i}, F_{\text{OSDA},i^*}),$$

where $N$ is the number of generated molecules, $F_{\text{gen},i}$ is the Morgan fingerprint of the $i$-th generated molecule, and $F_{\text{OSDA},i^*} = F_{\text{OSDA},j^*}$ with

$$j^* = \arg\max_{j} \text{sim}(F_{\text{gen},i}, F_{\text{OSDA},j}),$$

where $j$ iterates over all OSDAs, ensuring $F_{\text{OSDA},i^*}$ is the Morgan fingerprint of the OSDA most similar to the $i$-th generated molecule.

**MACCS.** MACCS fingerprints generate a fixed-length bit vector based on a predefined set of structural subgraphs or chemical features, with each bit representing the presence or absence of a specific chemical feature. Here, we use MACCS fingerprints to evaluate the difference between the generated molecules and the OSDAs reported in the literature. Specifically, this is done by calculating the

average similarity $S_{MACCS}$ between the MACCS fingerprints of the generated molecules and those of the closest OSDAs documented in the literature:

$$S_{MACCS} = \frac{1}{N} \sum_{i=1}^{N} \text{sim}(F_{\text{gen},i}, F_{\text{OSDA},i^*}),$$

where $N$ is the number of generated molecules, $F_{\text{gen},i}$ is the MACCS fingerprint of the $i$-th generated molecule, and $F_{\text{OSDA},i^*} = F_{\text{OSDA},j^*}$ with :

$$j^* = \arg \max_j \text{sim}(F_{\text{gen},i}, F_{\text{OSDA},j}),$$

with $j$ iterating over all OSDAs, ensuring $F_{\text{OSDA},i^*}$ is the MACCS fingerprint of the OSDA most similar to the $i$-th generated molecule.

**RDK.** RDK fingerprints generate feature bit vectors by traversing the bond paths in a molecule. Here, we use RDK fingerprints to evaluate the difference between the generated molecules and the OSDAs reported in the literature. Specifically, this is done by calculating the average similarity $S_{RDK}$ between the RDK fingerprints of the generated molecules and those of the closest OSDAs documented in the literature:

$$S_{RDK} = \frac{1}{N} \sum_{i=1}^{N} \text{sim}(F_{\text{gen},i}, F_{\text{OSDA},i^*}),$$

where $N$ is the number of generated molecules, $F_{\text{gen},i}$ is the RDK fingerprint of the $i$-th generated molecule, and $F_{\text{OSDA},i^*} = F_{\text{OSDA},j^*}$ with :

$$j^* = \arg \max_j \text{sim}(F_{\text{gen},i}, F_{\text{OSDA},j}),$$

with $j$ iterating over all OSDAs, ensuring $F_{\text{OSDA},i^*}$ is the RDK fingerprint of the OSDA most similar to the $i$-th generated molecule.

**WHIM.** WHIM descriptors are a class of molecular descriptors used to characterize the three-dimensional structure of molecules by considering global properties such as molecular geometry, mass distribution, polarity, and electron density. Here, we use WHIM descriptors to calculate the difference between the distribution of the generated molecules and the distribution of OSDAs reported in the literature. WHIM descriptors capture information about a molecule's three-dimensional conformation, including its size, shape, symmetry, and atomic distribution. Due to molecular flexibility, different conformations can result in significantly different WHIM representations. For example, a long linear molecule can either stretch out or fold, leading to two distinct 3D representations. To address this challenge, we calculated the average conformation WHIM descriptors using the geometries obtained from RDKit (Landrum, 2013). This approach allows us to capture the diverse 3D characteristics of each molecule across its various conformations.

**Energy Distance(ED).** Energy Distance is a statistical measure used to quantify the difference between two probability distributions. It is based on the concept of energy statistics and captures the distance between distributions in terms of expected Euclidean distances between samples. Energy Distance is symmetric and particularly sensitive to both location and shape differences between distributions. In this study, we compute the Energy Distance between the distribution of the generated molecules and the distribution of OSDAs reported in the literature, using the WHIM descriptors. The formula for Energy Distance is given by:

$$\text{ED}(X, Y) = 2 \cdot \mathbb{E}[d(X, Y)] - \mathbb{E}[d(X, X')] - \mathbb{E}[d(Y, Y')],$$

where $X$ and $Y$ represent independent samples from the generated molecules and OSDA distributions, respectively. Here, $d(X, Y)$ denotes the Euclidean distance between samples, and $\mathbb{E}$ denotes the expectation. The Energy Distance captures both the mean differences between the distributions and their internal variability, making it a robust measure for comparing the overall structure of the molecular distributions.

**KL-divergence.** Kullback-Leibler (KL) divergence is an asymmetric measure used to quantify the difference between two probability distributions. It measures the information loss or gain when one distribution is compared to another. Here, we use the WHIM descriptors of the molecules, after dimensionality reduction via PCA, to calculate the difference between the distribution of the generated molecules and the distribution of OSDAs reported in the literature. The formula for KL-divergence is:

$$D_{\mathrm{KL}}(P\|Q) = \int P(x) \log \frac{P(x)}{Q(x)}\, dx,$$

where $P(x)$ is the probability density function of the generated molecules' distribution, and $Q(x)$ is the probability density function of the OSDAs' distribution. The KL-divergence quantifies the amount of information lost when $Q$ is used to approximate $P$, with a larger value indicating greater divergence between the two distributions.

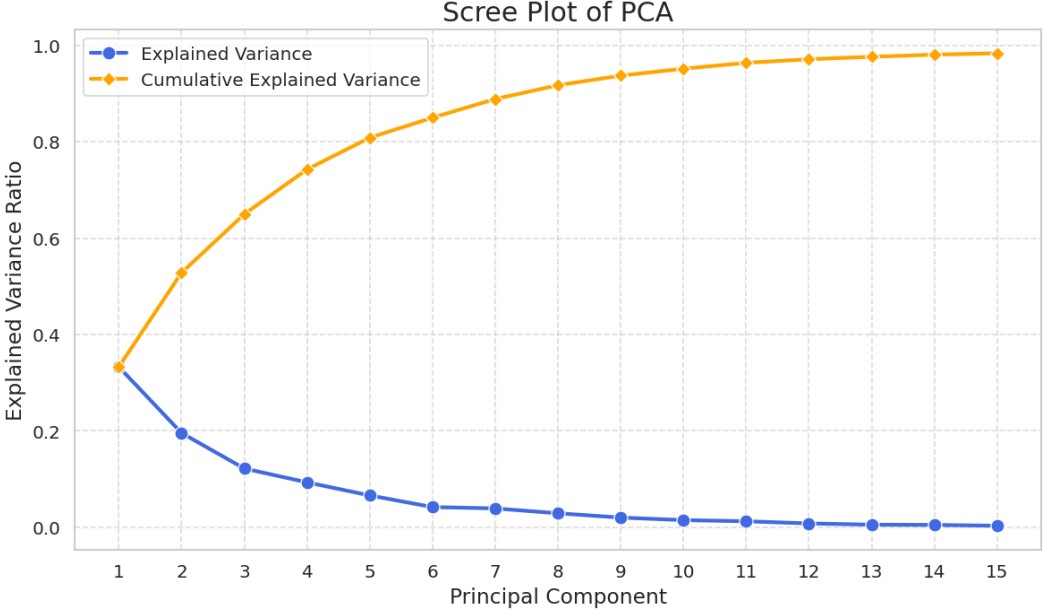

Figure 6: Scree plot of PCA on WHIM descriptors.

Table 2: Screening criteria

| Property | Criteria |
|---|---|
| Number of rotatable bonds | Less than 5 |
| Size of rings without bridge bonds | Smaller than 9-membered rings |
| Size of smallest rings | Larger than 4-membered rings |
| Distortion of N-rings | Must not contain triple bonds in N-rings |
| C to N ratio | More than 4 and less than 20 |
| Presence of $NH^+$ and $NH_3^+$ group | Must not be contained |
| Presence of unfavorable atoms and bonds | Must not contain these atoms and bonds (O, Si, P, S, B, F, $-O-C-O-$, $-C=O$, and $N-N$) |

# D  USING THE FEW-SHOT COT (CHAIN OF THOUGHT) PROMPT ENGINEERING METHOD FOR OSDA MOLECULE DESIGN CASES

In this section, we present the complete prompt that we used.

## Chain of Thought

**Let's think step by step** to design an effective OSDA molecule for a given zeolite structure. We will follow these steps:

**(1)Analyze the Target Zeolite Structure:** Begin by reviewing the key features of the target zeolite, including its pore size, topology, and framework type. Consider how these structural characteristics might influence the binding and stability of potential OSDA molecules.

**(2)Identify Functional Groups:** Based on the analysis of the zeolite structure, identify functional groups that are likely to enhance the interaction with the zeolite framework, Consider groups that can fit well within the pores and stabilize the structure.

**(3)Analyze Example Molecules:** I need to analyze the given examples to understand the relationship between the SMILES structures and their binding energies and synthesizability scores. This analysis will help guide the selection of structural features in the design of new OSDA molecules.

**(4)Generate SMILES for OSDA Candidates:** Using the identified functional groups, construct a series of SMILES representations for potential OSDA molecules. Ensure that these molecules are chemically valid and capable of directing the assembly of the zeolite structure.

Figure 7: Chain of Thought prompts in OSDA design.

## Chain of Thought prompts

**Relevant Context:** You are a chemist tasked with designing new Organic Structure Directing Agents (OSDAs) for AFI Zeolite. Your goal is to generate OSDAs with specific binding energies and synthesizability scores. Below are several examples, including the SMILES strings of the OSDAs, their binding energies, and synthesizability scores.
For instance, the SMILES string C[C@@H]1CCC[N+]12CCCCCC2 has a binding energy of -7.101 kcal/mol and a synthesizability score of 1.927.
Another example, NCCCNCCNCCCN, has a binding energy of -7.500 kcal/mol and a synthesizability score of 1.818.
The SMILES string C[N+]1(C)CC[C@@H]2CCCC[C@@H]2CC1 has a binding energy of -6.961 kcal/mol and a synthesizability score of 1.879.
Finally, the structure C[C@@H]1[C@@H]2CC[C@@](C)(CC[N+]2(C)C)[C@@H]1C has a binding energy of -8.199 kcal/mol and a synthesizability score of 2.544. Lower binding energies and synthesizability scores are preferred.

**Question:** Can you design a new OSDA of AFI Zeolite and provide its SMILES string? You just need to provide SMILES.

**Thought:** Let's think step by step to design an effective OSDA molecule for a given zeolite structure. We will follow these steps:
(1) Analyze the Target Zeolite Structure: Begin by reviewing the key features of the target zeolite, including its pore size, topology, and framework type. Consider how these structural characteristics might influence the binding and stability of potential OSDA molecules.
(2) Identify Functional Groups: Based on the analysis of the zeolite structure, identify functional groups that are likely to enhance the interaction with the zeolite framework. Consider groups that can fit well within the pores and stabilize the structure.
(3) Analyze Example Molecules: I need to analyze the given examples to understand the relationship between the SMILES structures, their binding energies, and synthesizability scores. This analysis will help guide the selection of structural features in the design of new OSDA molecules.
(4) Generate SMILES for OSDA Candidates: Using the identified functional groups, construct a series of SMILES representations for potential OSDA molecules. Ensure that these molecules are chemically valid and capable of directing the assembly of the zeolite structure.

**Action:** Finish[Cc1cc(C)[n+](C)n1C]

Figure 8: Details of the few-shot CoT (Chain of Thought) prompts. The data in the Relevant Context section comes from the OSDB database.

## Reflection prompts

**Relevant Context:** You are a chemist tasked with designing new Organic Structure Directing Agents (OSDAs) for AFI Zeolite. Your goal is to generate OSDAs with specific binding energies and synthesizability scores. Below are several examples, including the SMILES strings of the OSDAs, their binding energies, and synthesizability scores.
For instance, the SMILES string C[C@@H]1CCC[N+]12CCCCCC2 has a binding energy of -7.101 kcal/mol and a synthesizability score of 1.927.
Another example, NCCCNCCNCCCN, has a binding energy of -7.500 kcal/mol and a synthesizability score of 1.818.
The SMILES string C[N+]1(C)CC[C@@H]2CCCC[C@@H]2CC1 has a binding energy of -6.961 kcal/mol and a synthesizability score of 1.879.
Finally, the structure C[C@@H]1[C@@H]2CC[C@@](C)(CC[N+]2(C)C)[C@@H]1C has a binding energy of -8.199 kcal/mol and a synthesizability score of 2.544. Lower binding energies and synthesizability scores are preferred.

**Question:** Can you design a new OSDA of AFI Zeolite and provide its SMILES string? You just need to provide SMILES.

**Thought:** Let's think step by step to design an effective OSDA molecule for a given zeolite structure. We will follow these steps:
(1) Analyze the Target Zeolite Structure: Begin by reviewing the key features of the target zeolite, including its pore size, topology, and framework type. Consider how these structural characteristics might influence the binding and stability of potential OSDA molecules.
(2) Identify Functional Groups: Based on the analysis of the zeolite structure, identify functional groups that are likely to enhance the interaction with the zeolite framework. Consider groups that can fit well within the pores and stabilize the structure.
(3) Analyze Example Molecules: I need to analyze the given examples to understand the relationship between the SMILES structures, their binding energies, and synthesizability scores. This analysis will help guide the selection of structural features in the design of new OSDA molecules.
(4) Generate SMILES for OSDA Candidates: Using the identified functional groups, construct a series of SMILES representations for potential OSDA molecules. Ensure that these molecules are chemically valid and capable of directing the assembly of the zeolite structure.

**Action:** Finish[Cc1cc(C)[n+](C)n1N]

**Reflection:** My answer included a SMILES string that was not in the correct format. In the future, I should ensure that the SMILES strings are accurately represented and conform to standard notation to avoid similar issues.

Figure 9: Details of the reflection prompts. The data in the Relevant Context section comes from the OSDB database.

## Self-reflection

You have attempted to answer following question before and failed. The following reflection(s) give a plan to avoid failing to answer the question in the same way you did previously. Use them to improve your strategy of correctly answering the given question.

**Reflections:**
-My previous attempt to design a new OSDA for AFI zeolite failed because the synthesizability score of the generated SMILES string was too high. This indicates that the molecule was too difficult to synthesize. Additionally, the binding energy was not optimized to be as low as possible.
-My previous attempt failed because the generated SMILES string did not meet the empirical characteristics of OSDA molecules, specifically the nitrogen-to-carbon ratio and the structural requirements for rings and rotatable bonds.

## Actor

**Thought:**
Based on the reflections and the given context, it's clear that the previous attempts failed due to high synthesizability scores and not optimizing the binding energy. To avoid these issues, I need to design a molecule that is simpler to synthesize and has a lower binding energy. The given examples show that molecules with nitrogen and carbon atoms, as well as ring structures, tend to have lower binding energies and synthesizability scores. Therefore, I should aim to design a molecule with these characteristics. Additionally, I should aim to keep the molecule as simple as possible to ensure a lower synthesizability score.

Figure 10: Example of the reflection.

# E  BINDING ENERGY ESTIMATION MODEL DETAILS

This section covers the details and results of the Binding Energy Estimation Model, including our ablation study on the model.

The OSDA-Zeolite complexes in the real world are quite complex, as OSDAs may adopt multiple conformations when interacting with zeolites, resulting in complexes that do not exhibit the strict periodicity of typical crystals. However, both the complexes from the OSDB database and those generated from voids are stored as CIF files and treated as unit cells, as illustrated in Figure 11. Therefore, we continue to use crystal structures and periodic graph neural networks (CGCNN) for modeling in this context.

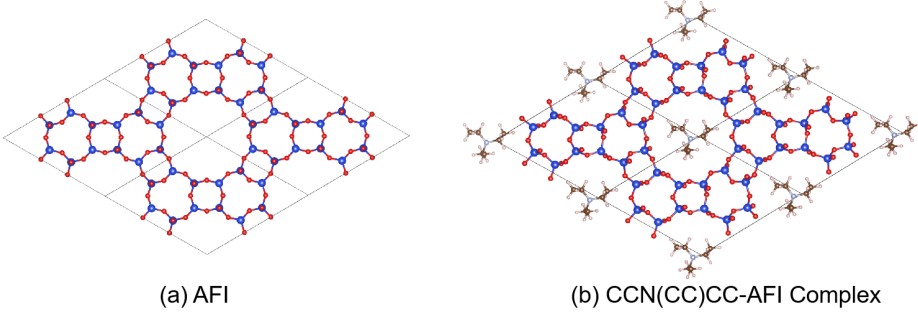

(a) AFI                          (b) CCN(CC)CC-AFI Complex

Figure 11: Dock illustration. (a) An AFI framework zeolite. (b) The combined complex with the 'CCN(CC)CC' OSDA.

Table 3: Binding energy prediction error (MAE) on the OSDB database.

| Model | Binding Energy (kJ/mol Si) (MAE ↓) |
|---|---|
| full model | **0.384** |
| Remove Zeolite encoder | 0.402 |
| Remove Smiles encoder | 0.411 |
| Only Complex encoder | 0.469 |

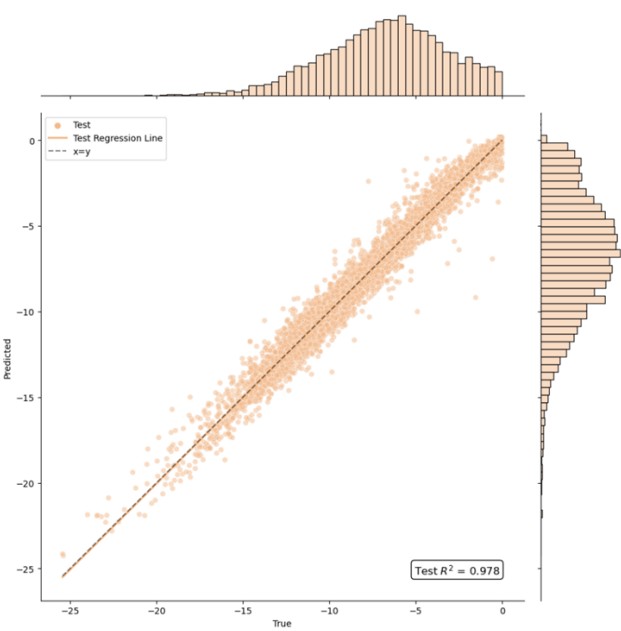

Figure 12: The results of the Binding Energy Estimation model on the test set.

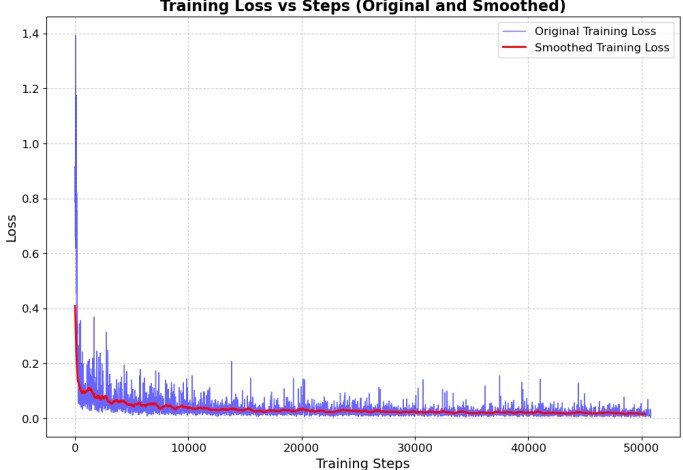

Figure 13: Model loss trends.

# F SUPPLEMENTARY EXPERIMENTAL RESULTS

This section primarily provides supplementary results from the experiments.

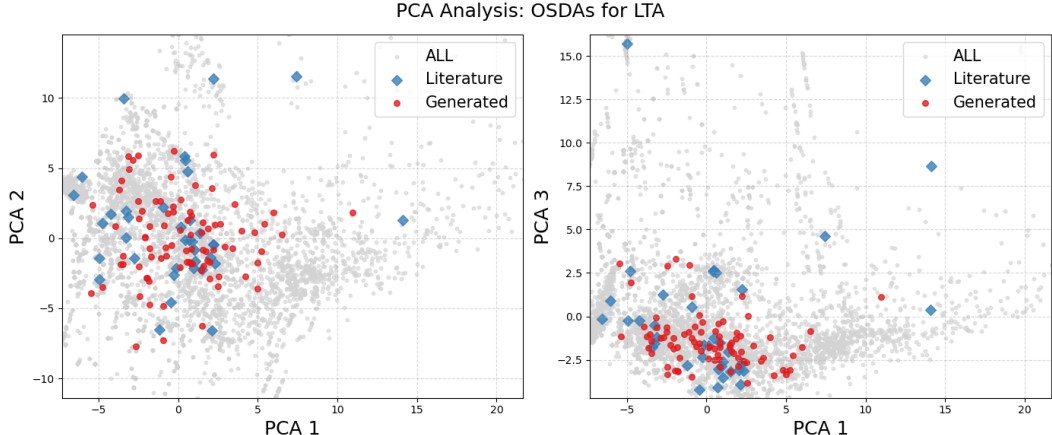

Figure 14: Principal Component Analysis (PCA) of WHIM vector representation for OSDA molecules used in LTA framework. PCA 1, 2, and 3 represent the first three principal axes. Red points show OSDAs generated by the OSDA Agent, blue points represent OSDAs of LTA framework from the literature, and gray points represent all OSDAs extracted from the literature.

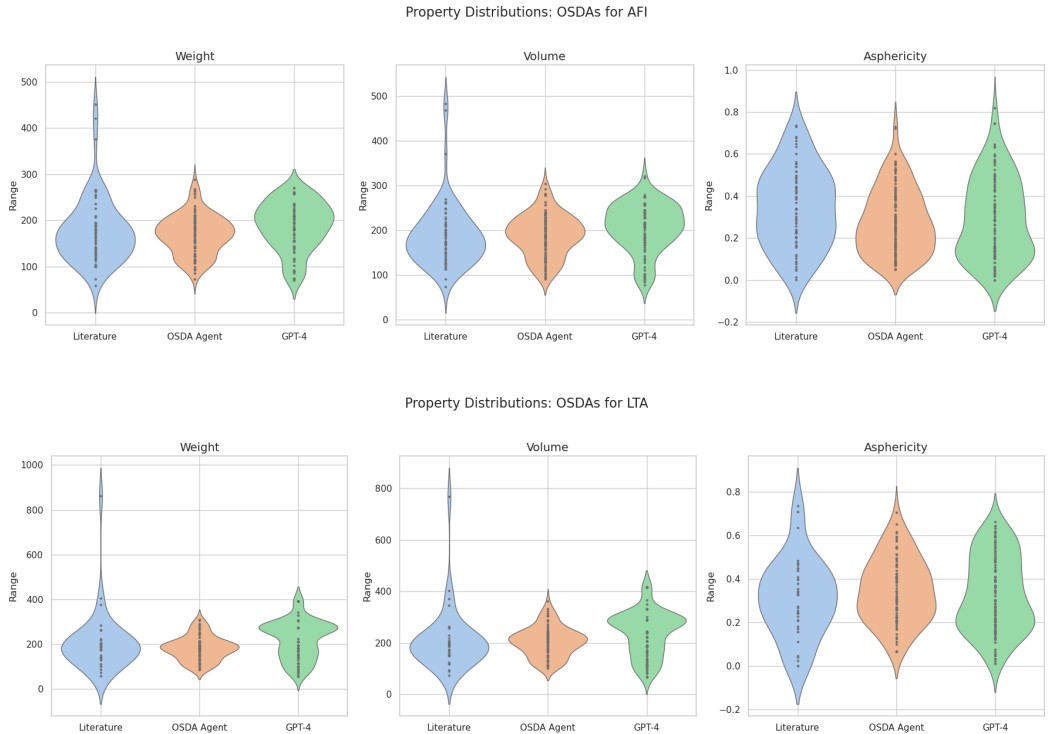

Figure 15: The distribution of molecular volume, molecular weight, and asphericity is compared between the OSDAs reported in the literature and those generated by the OSDA Agent in conjunction with GPT-4.

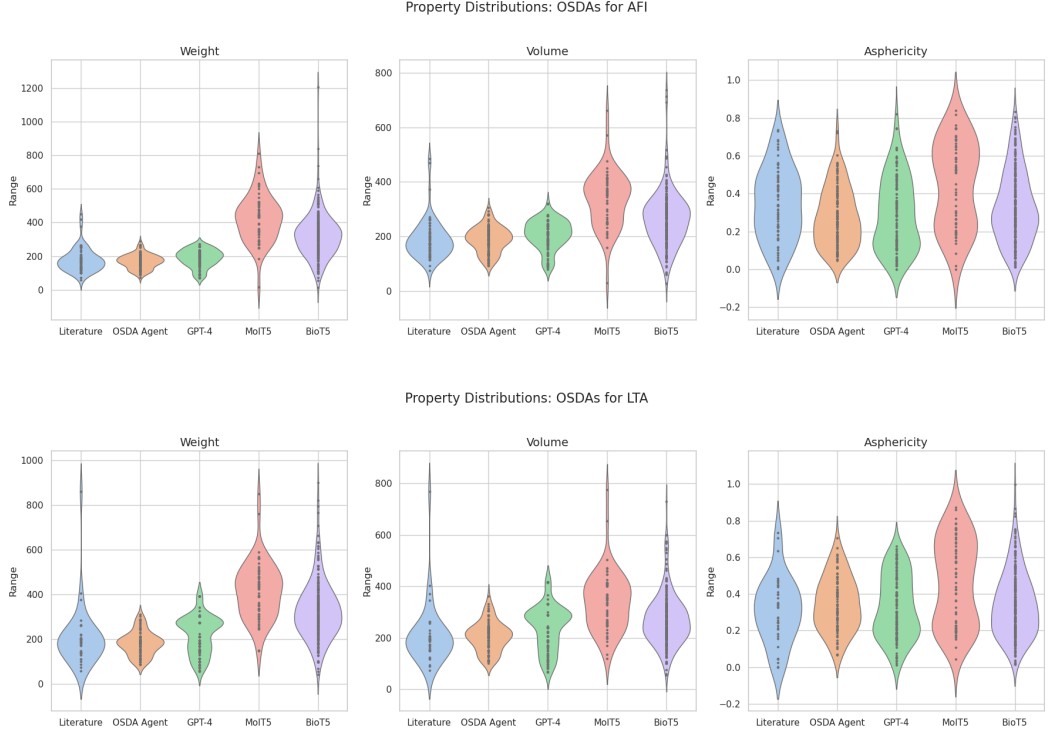

Figure 16: The distribution of molecular volume, molecular weight, and asphericity is compared between the OSDAs reported in the literature and those generated by the OSDA Agent in conjunction with GPT-4, BioT5, MolT5.

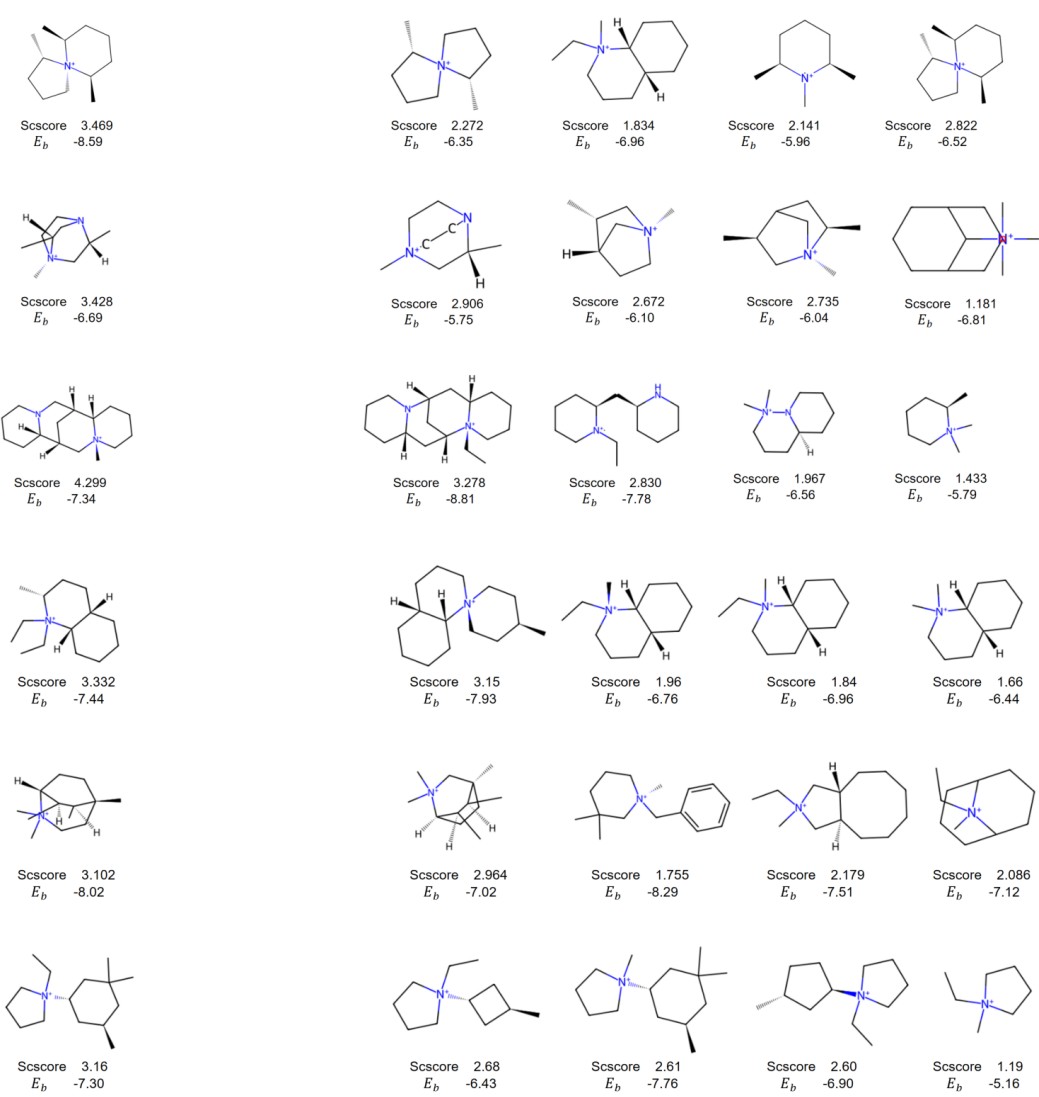

Figure 17: The optimization results of OSDA molecules are presented, with the OSDA molecules from the OSDB database on the left and the molecules optimized by the OSDA Agent on the right.

## G    ABLATION STUDY

In this section, we perform an ablation study on the proposed components, focusing on the reflection mechanism and the individual components within the Evaluator. To minimize randomness as much as possible, we use 15 identical sets of prompts to design OSDA molecules for AFI framework. The results obtained are as follows:

Table 4: Performance metrics for the ablation study of different methods.

| Method | Validity ↑ | BLEU ↑ | Morgan↑ | MACCS ↑ | RDK ↑ | ED ↓ | KL Divergence ↓ | Avg Rank |
|---|---|---|---|---|---|---|---|---|
| OSDA Agent | **1.000** | 0.601 | 0.368 | **0.816** | **0.623** | **0.934** | **0.825** | 1.28 |
| w/o reflection mechanism | 0.702 | 0.581 | 0.331 | 0.782 | 0.553 | 1.359 | 0.973 | 4.75 |
| w/o RDKit | 0.770 | 0.593 | 0.355 | 0.751 | 0.566 | 1.233 | 0.830 | 3.42 |
| w/o SCSore | **1.000** | 0.570 | **0.371** | 0.802 | 0.614 | 1.256 | 1.001 | 2.85 |
| w/o blending energy | **1.000** | **0.627** | 0.356 | 0.787 | 0.619 | 1.275 | 0.972 | 2.42 |

Here, we observe that removing the reflection mechanism or any individual component within the Evaluator leads to worse results.

## H  EVALUATION OF ALTERNATIVE LLM COMPONENTS

In this section, we explore the use of alternative large language models (LLMs) and compare their performance. Specifically, we tested two popular open-source LLMs, Mistral (Jiang et al., 2023) and LLaMA 3.1(Dubey et al., 2024), and observed that our method also resulted in substantial improvements in their ability to design OSDAs:

Table 5: Performance metrics for different methods.

| Method | BLEU ↑ | Morgan ↑ | MACCS ↑ | RDK ↑ |
|---|---|---|---|---|
| OSDA Agent | 0.601 | 0.368 | 0.816 | 0.624 |
| OSDA Agent* | 0.571 | 0.317 | 0.772 | 0.601 |
| llama | 0.522 | 0.301 | 0.628 | 0.416 |
| OSDA Agent(llama) | 0.551 | 0.315 | 0.754 | 0.565 |
| Mistral | 0.338 | 0.177 | 0.411 | 0.224 |
| OSDA Agent(Mistral) | 0.512 | 0.306 | 0.740 | 0.541 |

The OSDA Agent is our default model. The OSDA Agent* replaces the actor with GPT-4o while the OSDA Agent (Llama) is fully built on Llama, the OSDA Agent (Mistra) is fully built on Mistra. Our experimental results show that, regardless of the model used, our OSDA Agent significantly enhances the overall design results. However, currently, its performance is still slightly inferior to GPT-4, which remains the most effective model for this specific task.

## I  EXAMPLES OF OSDAS FOR AFI, AEI, LEV, AFX, MOR, LTA AND ITE ZEOLITES GENERATED USING THE OSDA AGENT

We used the OSDA Agent to generate OSDA molecules for seven types of zeolites, including AFI, AEI, LEV, AFX, MOR, LTA, and ITE. After consulting experts in materials and chemistry, they confirmed that the generated molecules have potential as OSDAs.

Table 6: Overview of the zeolites utilized.

| Framework Type | Crystal system | Length a (Å) | Length b (Å) | Length c (Å) | Angle $\alpha$ (°) | Angle $\beta$ (°) | Angle $\gamma$ (°) | pore size (Å) |
|---|---|---|---|---|---|---|---|---|
| LTA | Cubic | 11.92 | 11.92 | 11.92 | 90 | 90 | 90 | 4.21 |
| AFI | Hexagonal | 13.83 | 13.83 | 8.58 | 90 | 90 | 120 | 7.42 |
| ITE | Orthorhombic | 20.75 | 9.80 | 20.01 | 90 | 90 | 90 | 4.21 |
| LEV | Trigonal | 13.17 | 13.17 | 22.58 | 90 | 90 | 120 | 3.53 |
| MOR | Orthorhombic | 18.26 | 20.53 | 7.54 | 90 | 90 | 90 | 6.45 |
| AFX | Hexagonal | 13.67 | 13.67 | 19.70 | 90 | 90 | 120 | 3.73 |
| AEI | Orthorhombic | 13.68 | 12.61 | 18.50 | 90 | 90 | 90 | 3.84 |

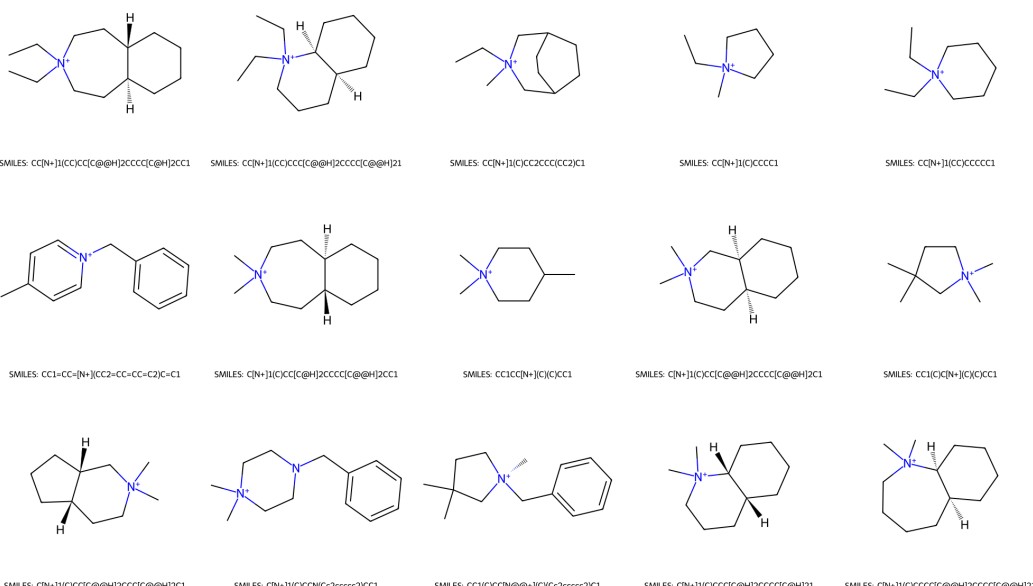

Figure 18: OSDAs for AFI framework

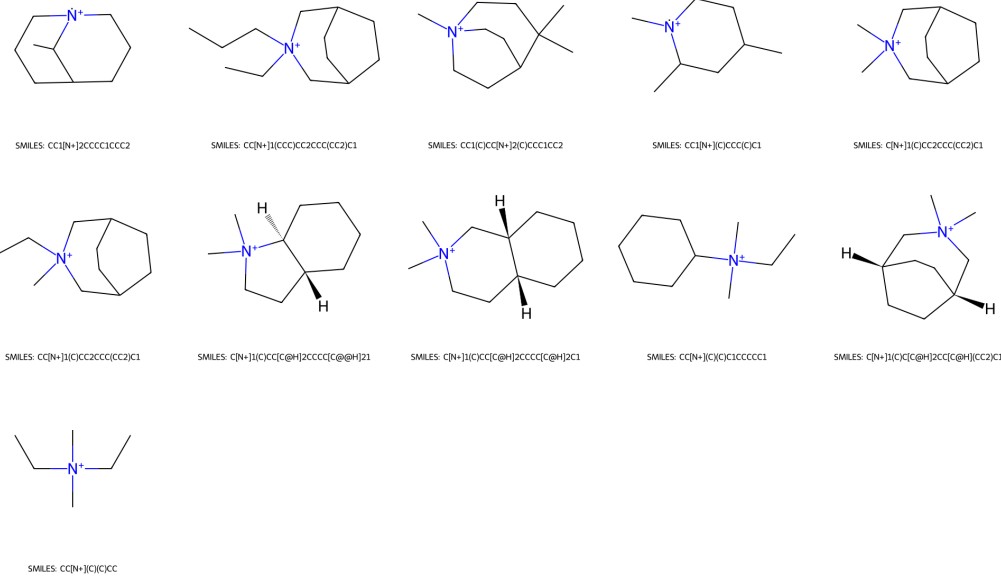

Figure 19: OSDAs for AEI framework

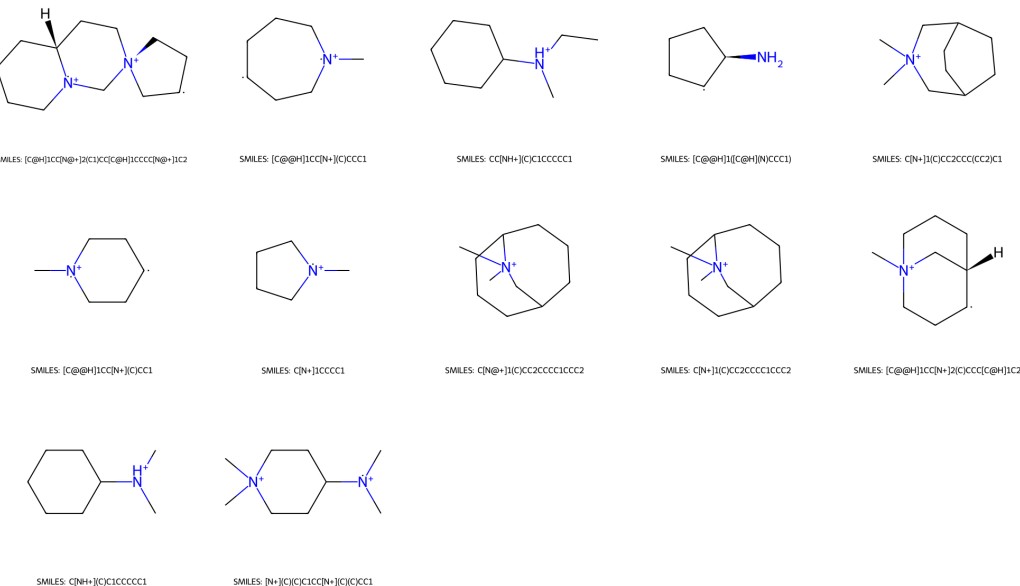

Figure 20: OSDAs for AFX framework

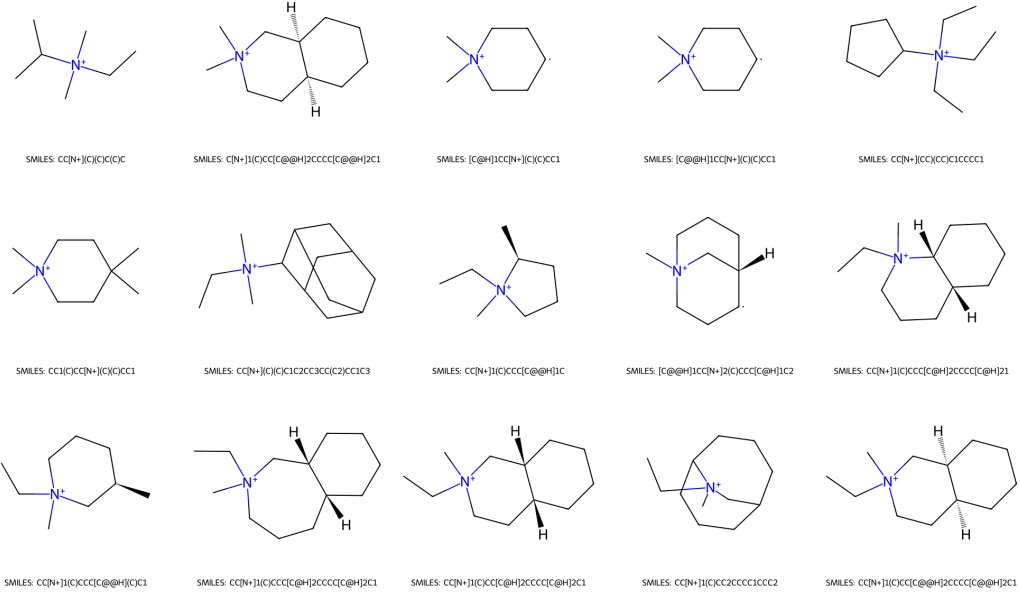

Figure 21: OSDAs for ITE framework

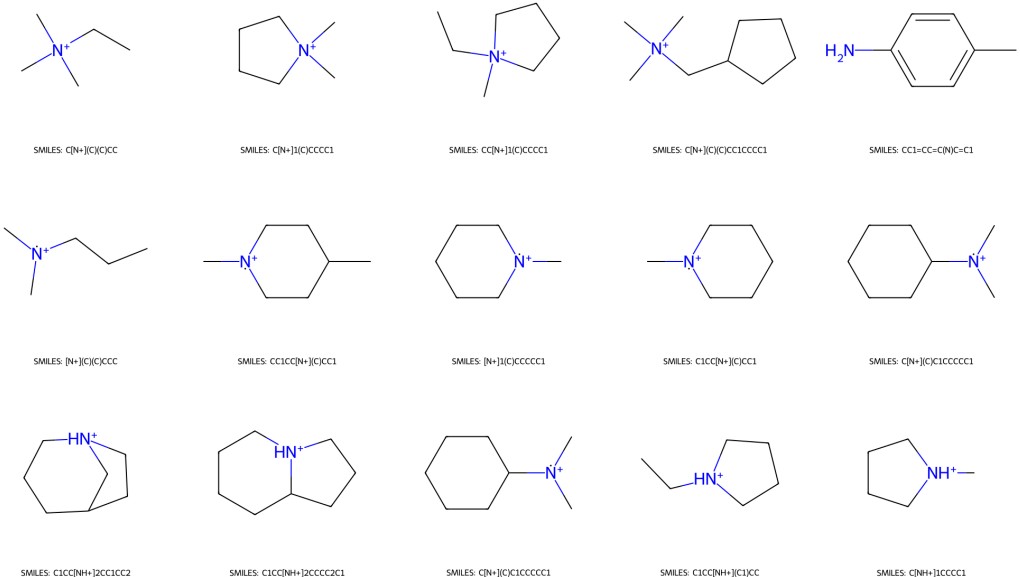

Figure 22: OSDAs for LEV framework

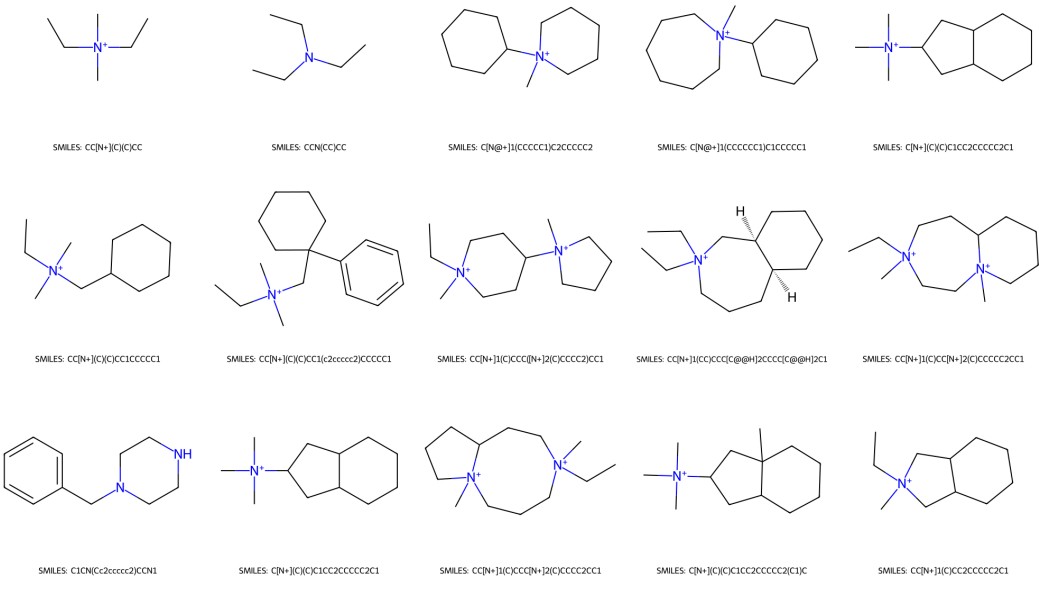

Figure 23: OSDAs for LTA framework

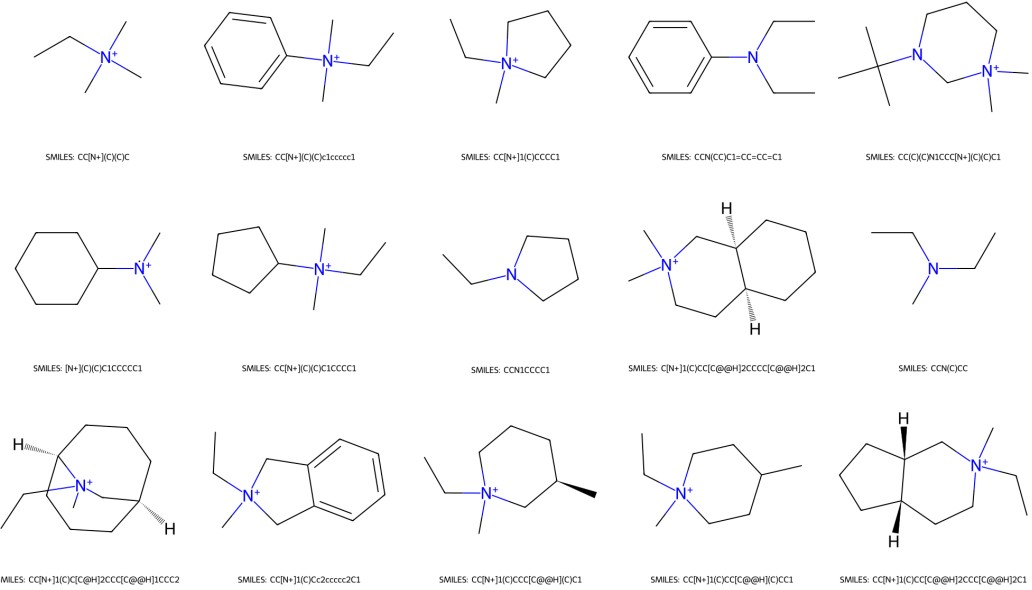

Figure 24: OSDAs for MOR framework

## J  DEFINITION AND EXPLANATION OF TERMS AND CONCEPTS IN THE PAPER

**SMILES (Simplified Molecular Input Line Entry System)**
SMILES is a symbolic language used to represent the structure of chemical molecules. It encodes the atoms, bonds, and spatial arrangements of a molecule in a string of characters. SMILES is widely used in computational chemistry and AI for Science due to its simplicity in storing and exchanging chemical information and its ability to be parsed by computer programs.

**In-Context Learning (ICL)**
In-Context Learning refers to the process where a pre-trained language model performs reasoning and generates responses based on provided examples or contextual information in the input. This is done without the need for additional model training or parameter updates, making it a form of immediate inference for a given task.

**Chain of Thought**
Chain of Thought refers to a reasoning process where a model breaks down a complex problem into a sequence of logical steps to improve the accuracy and transparency of its reasoning. This approach helps in step-by-step problem-solving by guiding the model through intermediate steps.

**RDKit**
RDKit is an open-source toolkit for cheminformatics that provides a wide range of functionalities for molecular manipulation, feature extraction, structure visualization, and drug design. It is widely used in chemistry and bioinformatics.

**SCScore (Synthetic Complexity Score)**
SCScore is a metric used to assess the synthetic difficulty of a chemical molecule. It considers factors such as the number of synthetic steps, reagents, and reaction conditions required to synthesize the molecule, evaluating its complexity from laboratory synthesis to industrial production. A higher score indicates greater synthetic difficulty.

**Zeolites**

Zeolites are a class of porous materials with a regular framework structure composed of silicon-oxygen ($SiO_4$) and aluminum-oxygen ($AlO_4$) tetrahedra linked by oxygen bridges. Their unique molecular sieve properties allow them to selectively adsorb and separate molecules based on size and shape. Zeolites are widely used in various industries for applications such as catalysis, adsorption, ion exchange, gas separation, and water treatment. Due to their tunable structure, the pore size and crystal morphology of zeolites can be optimized by using different organic structure-directing agents (OSDAs) to meet specific application requirements.

**OSDA**

OSDA (Organic Structure-Directing Agent) refers to an organic compound, typically a quaternary ammonium salt or other organic molecules, that is used in the synthesis of zeolites to guide the formation of their crystalline structure. During the synthesis process, OSDAs interact with the aluminosilicate framework and help direct the arrangement of atoms, thereby influencing key characteristics of the zeolite such as pore size, pore shape, and crystal morphology. The use of different OSDAs allows for the tuning of the zeolite's properties to suit specific industrial applications, such as catalysis, adsorption, and gas separation.

**VOID**

VOID (Voronoi Organic-Inorganic Docker) is a tool for molecular docking and materials design, especially for nanoporous materials. It uses Voronoi diagrams to identify docking sites in crystal structures and combines geometric fitness functions and Monte Carlo methods to optimize docking. Supporting batch docking with tensor operations for efficiency, VOID is widely used in materials science and molecular modeling.

