# OpenReview forum: "OSDA Agent: Leveraging Large Language Models for De Novo Design of Organic Structure Directing Agents"
_ICLR.cc/2025/Conference — ICLR 2025 Spotlight_

### Official Review · Reviewer_WsXS · 2024-10-21

**Soundness:** 3
**Presentation:** 2
**Contribution:** 2
**Rating:** 6
**Confidence:** 2

**Summary:**

The authors introduce OSDA Agent, an interactive framework for designing Organic Structure Directing Agents (OSDAs) used in zeolite synthesis. The framework leverages large language models (LLMs) as the core intelligence, complemented by computational chemistry tools. OSDA Agent consists of three key components: the Actor (generates potential OSDA structures), the Evaluator (assesses generated OSDAs using computational tools), and the Self-reflector (produces reflective summaries to refine subsequent outputs).
The main advantages of OSDA Agent are:

- Improved generation quality compared to pure LLM models, producing candidates consistent with experimentally validated OSDAs.
- Integration of chemical knowledge and tools to ensure generated molecules adhere to chemical rules and are feasible.
- Interactive and iterative design process that leverages feedback and self-reflection

Experiments demonstrate that OSDA Agent outperforms baseline methods, including state-of-the-art text-based de novo molecule generation approaches.

**Strengths:**

[Algorithm use] The work seems to highlight the successful use of existing LLM pipelines in searching a vast space of objects of scientific discovery. In this case, the objects were chemical agents, OSDAs, but theoretically, these could also be molecules, drugs, equations, etc. It also presents the possibility of connecting LLM pipelines to more traditional algorithms or solutions which allow evaluating the outputs of LLM models and sending feedback to improve future iterations. This is valuable for the community since it shows the viability of prior work.

[Results] Building on the previous strength, the use of the existing methods allowed authors to discover researched OSDAs as well as potentially new ones. This shows the robustness of the framework, possibly opening new avenues to zeolite synthesis. The real consequences of the introduced method are difficult to predict without expert knowledge of the chosen topic.

**Weaknesses:**

[Motivation] Being a non-expert in chemistry it is difficult to grasp why OSDAs pose such a challenge to contemporary methods. Linked to that, the description of related work could use some more detail, at least 1 example of an existing method and the way it works with a discussion on how the method presented in this work builds upon its weaknesses. The authors do try to convey that new algorithms for OSDAs are needed but it isn’t clear to me why. Is it only the need to give feedback to the algorithm in order to get better outputs with the next iteration (impossible to achieve with traditional ML methods)? If so, why would it be so crucial for OSDAs? Is it that the current methods are ineffective for some subclass of OSDAs or…?

[Specificity] The work may be certainly worthwhile to OSDA specialists but I have doubts whether it would be of general interest. It shows how to apply an existing framework to a specific task of chemical agent construction. There seems to be limited novelty when it comes to representation learning itself or algorithmics in general. The reflection mechanism that verbalizes errors found by the Evaluator for future learning seems to be taken from a paper by Shin et al. (2024), and adjusted for this particular task, similarly to other used algorithms.

[Clarity] As much as the manuscript is written well and it is possible to understand the syntax and often what is being conveyed, some of the jargon becomes weary and makes details intelligible. This happens in Section 3, where the data is described and seemingly one of the crucial contributions of the work — the binding energy. It is also evident in Section 4.1 —the authors talk about SMILES sequences, In-Context Learning, Chain of Thought, etc. without ever defining, even intuitively, what these acronyms convey. See also RDKit, SCScore, etc. Section 4.1. would benefit from explaining intuition on the framework rather than an overview with specifics defined later.

[Details] There are a lot of moving parts in the whole framework but there is little detail on how they work exactly. Which of the parts are new knowledge? How does the whole framework operate, from start to finish? Each step in the processing could be described in a separate subsection, to streamline presentation.

**Questions:**

1. Can you provide more context on why OSDAs pose such a challenge to contemporary methods?
2. What specific aspects of OSDA design call for an interactive, feedback-driven approach like the one you've developed? Are there certain subclasses of OSDAs that current methods struggle with?
3. Can you provide an overview of the working of the whole OSDA framework descriptively, without pointing to existing work?
4. How does this work contribute to the broader fields of representation learning or algorithmic design beyond an application of existing methods to OSDA synthesis? Can you elaborate on the novelty of your approach compared to existing frameworks?
5. How does the binding energy estimation model work, and why is it crucial for your approach? Could you explain this concept in more accessible terms?

---

> ### Author Response · Authors · 2024-11-20
>
> ## (1/4)
>
> Thank you for recognizing the **novelty** of our work in [Algorithm use] and the **effectiveness** of our [results]. We also appreciate your constructive feedback.
>
> Thank you for your thoughtful comments. We appreciate your feedback, and we will address each of your concerns one by one.
>
> # The Importance of OSDA in Zeolite Synthesis
>
> Organic Structure-Directing Agents (OSDAs) are essential in zeolite synthesis, guiding the formation of their unique porous structures. Zeolites, with their selective molecular sieving properties, are used in catalysis, adsorption, ion exchange, and gas separation. OSDAs interact with the aluminosilicate framework during synthesis to control pore size, diameter, and crystal morphology, influencing zeolite functionality. The choice of OSDA is key to tailoring zeolite properties for specific industrial and environmental applications.
>
> # Why OSDAs pose a challenge to contemporary methods？
>
> （Weaknesses[Motivation] and question 1）
>
> OSDA plays a crucial role in determining the topology of zeolites, especially in the formation of their porous structures. Traditional methods mainly rely on two approaches: first, using empirical experience and experimentation to search for feasible OSDA molecules within a molecular library; and second, employing computational simulations of zeolite-OSDA interactions (such as Density Functional Theory, DFT) to assist in OSDA design. Both experience-based and simulation-based traditional methods are highly time-consuming and resource-intensive. In particular, the vast chemical space of OSDA (small molecules) presents significant challenges in discovering new OSDA candidates using conventional methods.
>
> Moreover, due to the difficulty in finding new OSDA molecules, the existing pool of OSDA candidates is **very limited**. For example, the Jensen dataset, which collects data from papers published between 1966 and 2020, contains only **758** different OSDA molecules. Traditional machine learning generation methods typically require large datasets, but the small number of OSDA molecules makes it challenging to train effective generative models. Several traditional generative model approaches[1,2] are typically trained using synthetic routes. This means that when designing OSDAs , information related to the synthetic pathway, such as gel chemistry, must be provided. Moreover, the limited number of different OSDA molecules in the dataset negatively impacts the model's ability to search effectively within the chemical space. Currently, Large Language Models (LLMs) have broad knowledge across related fields, and we aim to leverage the domain expertise of LLMs to overcome the data scarcity issue.
>
> # Why do we use an interactive, feedback-driven approach?
>
> （Weaknesses[Motivation] and question2）
>
> Due to the phenomenon of "hallucination" in large language models (LLMs), especially when performing complex tasks like molecular design, it is necessary to introduce additional chemical knowledge to help the LLMs accomplish the task more effectively. Through an interactive, feedback-driven approach, we professionally evaluate the molecules designed by the LLM, including factors such as OSDA empirical rules, molecular synthesis difficulty, safety, etc., and provide feedback to the LLM. This process is similar to how chemists validate through experiments and make improvements based on the experimental results.
>
> [1] Jensen Z, Kwon S, Schwalbe-Koda D, et al. Discovering relationships between OSDAs and zeolites through data mining and generative neural networks[J]. ACS central science, 2021, 7(5): 858-867.
>
> [2] Xu L, Peng X, Xi Z, et al. Predicting organic structures directing agents for zeolites with conditional deep learning generative model[J]. Chemical Engineering Science, 2023, 282: 119188.

---

> ### Author Response · Authors · 2024-11-20
>
> ## (2/4)
>
> # The Novelty of Our Method
>
> （Weaknesses[Clarity] and question 4）
>
> **About the Motivation of the Work**
>
> Our motivation for using an LLM Agent in OSDA molecule design stems from the challenge of **limited data** availability in this field (with only 758 distinct OSDA molecules in the dataset), which makes traditional machine learning methods difficult to train. Our approach leverages the extensive knowledge of LLMs, supported by specialized chemical tools, and employs a design-evaluation-reflection-improvement paradigm to enhance performance in OSDA molecule tasks.
>
> Moreover, our method is highly adaptable to other similar chemical tasks. Thanks to the flexibility of our framework, new chemical tools and reflection mechanisms can be customized to meet the specific needs of different tasks. This makes our approach a viable solution for many data-scarce generative tasks in the AI for Chemistry domain.
>
> **Algorithmic Innovations**
>
> While we did draw inspiration from the work of Shin et al. (2024), the reflection mechanism in our study goes beyond simple adjustments and has been customized extensively for chemical generation tasks. Our reflection mechanism dynamically integrates feedback from multiple complex chemical evaluation tools (e.g., SCScore, binding energy estimation).We have also demonstrated the effectiveness of the proposed reflection mechanism through ablation studies.
>
> | Method                     | Validity $\uparrow$ | BLEU $\uparrow$ | Morgan $\uparrow$ | MACCS $\uparrow$ | RDK $\uparrow$ | ED $\downarrow$ | KL Divergence $\downarrow$ | Avg Rank |
> |----------------------------|---------------------|-----------------|-------------------|------------------|----------------|-----------------|----------------------------|----------|
> | OSDA Agent                 | **1.000**              | 0.601         | 0.368            | **0.816**           | **0.624**         | **0.934**          | **0.825**                     | **1.28**     |
> | Remove reflection mechanism | 0.702              | 0.581          | 0.331            | 0.782           | 0.553         | 1.359          | 0.973                     | 4.57     |
> | Remove RDKit                | 0.770             | 0.593          | 0.355            | 0.751         | 0.566         | 1.233          | 0.8230                     | 3.42     |
> | Remove Scscore              | 1.000              | 0.570          | **0.372**            | 0.802           | 0.614        | 1.256          | 1.001                     | 2.85     |
> | Remove blending energy      | 1.000              | **0.627**          | 0.356            | 0.787           | 0.619         | 1.275          | 0.972                     | 2.42     |
>
> The OSDA Agent is our default model, and removing the reflection mechanism or any individual component within the Evaluator results in worse performance. Through the *design-evaluation-reflection-improvement* paradigm, we significantly enhanced the performance of OSDA molecule generation and provided a generalizable solution for other data-scarce generative tasks in AI for Chemistry.
>
> **Representation Learning Innovations**
>
> Our approach introduces a novel representation method for binding energy estimation. While prior studies have explored the representation of crystals, zeolites, and materials, very few have investigated the representation of complexes involving organic molecules and these materials. Most previous research on complexes has focused on organic macromolecules and small molecules, such as proteins.
>
> To the best of our knowledge, the only study on Organic-Inorganic complex representation is our concurrent work, Zeoformer[1], which focuses on the coarse-grained periodicity of OSDA-zeolite complexes. In contrast, our binding energy estimation model starts from the definition of binding energy and not only extracts features from the OSDA-zeolite complex but also incorporates distinct features of both OSDA and zeolite.
>
> | Model                      | Binding Energy (kJ/mol Si) (MAE ↓) |
> |----------------------------|------------------------------------|
> |  Only Complex encoder               |        0.469                  |
> |   Complex encoder+Zeolite encoder    |        0.411                     |
> | Complex encoder+ Smiles encoder      |       0.402                       |
> | full model       | **0.384**                             |
>
> The Complex encoder, Zeolite encoder, and Smiles encoder respectively represent the extraction of complex information, zeolite framework information, and molecular information.
> This novel representation method enhances the accuracy and relevance of binding energy estimation and advances the study of Organic-Inorganic complexes.
>
> [1]Shen X, Wan Z, Wen L, et al. Zeoformer: Coarse-Grained Periodic Graph Transformer for OSDA-Zeolite Affinity Prediction[J]. arXiv preprint arXiv:2408.12984, 2024.

---

> ### Author Response · Authors · 2024-11-20
>
> ## (3/4)
>
> # Definition of terms
>
> （weakness [Clarity]）
>
> Thank you for your feedback. We will clarify the meanings of the terms and acronyms used in the manuscript both in the main text and the appendix to improve readability.
>
> Here are the definitions of some key terms：
>
> **SMILES (Simplified Molecular Input Line Entry System)**
>
> SMILES is a symbolic language used to represent the structure of chemical molecules. It encodes the atoms, bonds, and spatial arrangements of a molecule in a string of characters. SMILES is widely used in computational chemistry and AI for Science due to its simplicity in storing and exchanging chemical information and its ability to be parsed by computer programs.
>
> **In-Context Learning (ICL)**
>
> In-Context Learning refers to the process where a pre-trained language model performs reasoning and generates responses based on provided examples or contextual information in the input. This is done without the need for additional model training or parameter updates, making it a form of immediate inference for a given task.
>
> **Chain of Thought**
>
> Chain of Thought refers to a reasoning process where a model breaks down a complex problem into a sequence of logical steps to improve the accuracy and transparency of its reasoning. This approach helps in step-by-step problem-solving by guiding the model through intermediate steps.
>
> **RDKit**
>
> RDKit is an open-source toolkit for cheminformatics that provides a wide range of functionalities for molecular manipulation, feature extraction, structure visualization, and drug design. It is widely used in chemistry and bioinformatics.
>
>
> **SCScore (Synthetic Complexity Score)**
>
> SCScore is a metric used to assess the synthetic difficulty of a chemical molecule. It considers factors such as the number of synthetic steps, reagents, and reaction conditions required to synthesize the molecule, evaluating its complexity from laboratory synthesis to industrial production. A higher score indicates greater synthetic difficulty.
>
> **Binding Energy**
>
> Binding energy refers to the interaction energy between an OSDA (organic structure-directing agent) molecule and the active site in a zeolite lattice. It is used to describe and predict the stability of the OSDA during the zeolite synthesis process.

---

> ### Author Response · Authors · 2024-11-20
>
> ## (4/4)
>
> # Details of Our Method
> （weakness[Details] and question 3,5）
>
> ## OSDA Agent framework
>
> Our OSDA agent framework is designed for optimizing the OSDA (Organic Structure-Directing Agent) molecules for zeolite design or improving an existing molecule. The simplified process works as follows:
> **Task Input**: When we need to design an OSDA for a specific zeolite or optimize an existing molecule, we first provide the task requirements through a carefully designed prompt. This prompt is input into a large language model (LLM), referred to as the Actor. Based on the task requirements, the LLM generates an initial OSDA molecule.
>
> **Molecule Evaluation**: The generated molecule is then fed into an evaluation module, called EVALUATION, which consists of several chemical tools:
>
> 1.RDKit Screening Criteria: Based on chemical expert experience, we have designed a set of screening standards to verify and assess the generated molecule's feasibility and validity.
>
> 2.SCScore (Synthesis Complexity Score): This score evaluates the synthetic difficulty of the molecule, considering factors such as the number of synthetic steps, reagents required, and the overall feasibility of synthesis.
>
> 3. Binding Energy Estimation Model: This model uses multiple information fusion techniques to estimate the binding energy between the OSDA molecule and the active sites in the zeolite lattice, which is crucial for predicting stability during the zeolite synthesis process.
>
> **Feedback and Self-Reflection**: The evaluation results are then provided to another large language model, called self-reflection, which summarizes the evaluation feedback. This model incorporates the results with expert knowledge to generate suggestions for improving the molecule’s design.
>
> **Final Optimization**: The feedback is sent back to the LLM (Actor), which uses it to refine the OSDA molecule further, ensuring it meets the synthesis requirements, adheres to expert rules, and has the desired binding energy.
>
> Through this iterative process, the framework is able to generate OSDA molecules that are optimized for synthesis feasibility, adhere to empirical rules, and exhibit the ideal binding energy for zeolite synthesis
>
> ## Binding energy estimation mode
>
> **Importance of Binding Energy**
>
> In zeolite synthesis, the organic structure-directing agent (OSDA) interacts with zeolite precursors, guiding the formation of a specific porous structure. The compatibility between OSDA and the zeolite is critical, as it determines whether the OSDA can effectively facilitate the crystallization process. This compatibility is often reflected in the binding energy: lower binding energy usually indicates stronger interactions between the OSDA and zeolite, making the OSDA more likely to succeed in synthesizing that zeolite type. Empirical evidence also shows that discovered OSDAs for specific zeolites tend to exhibit lower binding energies.
>
> **How the Model Works**
>
> Our binding energy estimation model is grounded in the fundamental formula of binding energy, which depends on the interactions among three components: the OSDA-zeolite complex, the zeolite itself, and the OSDA molecule. Here's how we estimate binding energy:
>
> **Generating the Complex:**
>
> We use VOID to simulate multiple docking poses of the OSDA within the zeolite framework. The pose with the lowest energy is selected as the estimated binding energy.
>
> **Extracting Features:**
>
> Complex Features: A crystal graph convolutional neural network (CGCNN) is employed to extract structural and energetic features of the docked OSDA-zeolite complex.
>
> Zeolite Features: Another CGCNN captures the standalone properties of the zeolite framework.
>
> OSDA Features: A pre-trained chemical transformer model extracts molecular-level features of the OSDA.
>
> **Feature Fusion and Prediction:**
>
> The features from these three components are fused to estimate the binding energy accurately.
> This integrated approach ensures the estimation considers all relevant interactions, making it a robust tool for predicting OSDA compatibility with specific zeolites
>
> We greatly appreciate your valuable feedback on our manuscript and acknowledge the areas that require improvement. We are committed to addressing the issues you raised during the camera-ready revision phase and will ensure that both clarity and detail are significantly enhanced.

---

> ### Comment · Reviewer_WsXS · 2024-11-23
> **Thank you for the responses but please include some of the explanations in the main paper**
>
> Thank you for the very detailed responses. I now better understand the main contribution of the paper. I agree that it is relevant. Please, make sure that you do convey the explanation why generating OSDAs poses a challenge to existing methods (including traditional ML) in the main paper, it much better underscore the importance of the introduced method.
>
> I would also suggest to the OSDA agent framework the same way you did in the response in the main text. It makes the framework clear. It also shows that the framework has the potential to generalize. Your contribution on Binding energy estimation would benefit if added to the appendix.
>
> I changed my score to Above acceptance threshold. I would score it higher, but I cannot evaluate if the Molecule Evaluation part of the framework is generalizable. Overall, like I said above and like you indirectly convey in the paper and your replies, the framework does have the potential to be used in other limited-data use-cases. I can see how it would work in other chemistry-related topics. But whether you could incorporate Evaluation from entirely different area, and still harness the power of LLMs during Feedback and Self Reflection is unclear. If it is clear to you, please openly spell it out with your reasoning in the paper.

---

> ### Author Response · Authors · 2024-11-25
> **Response to Reviewer WsXS**
>
> Dear Reviewer WsXS,
>
> Thank you very much for your detailed feedback and for taking the time to assess our work carefully. We truly appreciate your constructive suggestions and the recognition of the relevance of our contributions.
>
> Since large language models (LLMs) are pre-trained on extensive corpora, we believe they have the ability to handle other similar tasks. Although the current research focuses specifically on OSDA design for zeolites, we are confident that the underlying methods can be extended to other types of molecular design. The model is based on large language models that have been trained on various scientific texts, providing foundational chemical knowledge that supports broader molecular design capabilities. Moreover, the LLM Agent offers high flexibility and adaptability, enabling it to choose appropriate tools and evaluation methods based on specific design requirements. This adaptability allows the model to be applied to a wider range of materials and chemical tasks beyond zeolites and OSDA. We plan to further explore applying our approach to a broader range of chemical design tasks in the future.
>
> We take your comments very seriously and are committed to addressing them thoroughly in the camera-ready version.

---

### Official Review · Reviewer_t8Dz · 2024-11-03

**Soundness:** 2
**Presentation:** 3
**Contribution:** 3
**Rating:** 8
**Confidence:** 3

**Summary:**

This paper introduces the OSDA Agent, an interactive framework specifically made for designing organic structure directing (OSDA) agents, which is critical for zeolite synthesis. The main goal of the OSDA agent is to generate the de novo design of OSDAs with given zeolites as the target. This framework integrates computational chemistry tools into LLMs to check the quality of generated molecules and includes a reflection mechanism to improve OSDA generation. The OSDA Agent proposed in this work contains three key components: actor, evaluator, and self-reflector. This novel framework design creates a continuous learning environment that would facilitate more efficient and effective zeolite synthesis planning. Experimental results from the paper demonstrate the OSDA Agent yields better quality results compared to existing baseline models.

**Strengths:**

1. Novel chemistry LLM agent for molecule generation with a reflection mechanism design: the self-reflection modules in the OSDA Agent reuse feedback from previous iterations to optimize future trials, and this enhances the agent's decision-making ability effectively.
2. Tools to evaluate the quality of generated molecules: this model also includes three chemical tools to check if the generated chemical is valid and feasible, by checking the validity with RDKit, synthetic complexity. To estimate the binding energy, the authors also train their own binding energy estimation model that has lower computational complexity compared to existing computational methods.

**Weaknesses:**

1. Limited post-generation check: this agent framework includes three chemical property checks, and those would be useful for checking if the generated chemical is valid or reasonable, however, it does not consider other properties such as toxicity and explosiveness, the safety of the chemical in general.
2. see questions below

**Questions:**

1. Robustness of the input to the agent model: as there are multiple ways to represent a chemical, does this model support multiple chemical expressions, i.e. chemical formulas like H2O and IUPAC names like oxidane? The 'Evaluator' in the agent model would convert the input target zeolite into a SMILES string, but have you considered changing the chemical expression to see if the 'Evaluator' still works?
2. Accuracy of binding energy estimation model: this work proposes a new binding energy estimation model to lower the computational complexity of previous computational tools, but how accurate is this estimation model, and is the estimation from this model comparable to traditional atomic simulation methods?
3. LLM consideration: this work considers OpenAI's GPT models as the base LLM for the agent, and they are not open-source models and would be potentially costly to use, is there any reason that the authors did not also consider open-source models like Llama and Mistral, or would you consider experimenting with those open-source models?
4. Performances of the baseline models: curious about performance compared to baseline models, specifically for the 'validity' metric the authors proposed, how is this calculated, is it based on results from multiple computational models? This question stems from the result showing that one of the baseline models has a validity score of 0.00 for both datasets whereas the proposed model has a score of 100.
5. Memory and cost: this agent model includes a reflection mechanism and the paper also shows the number of reflections vs SCScore plot, showing that 4 reflections would reduce the synthesis difficulty, would having more reflections yield even better performances, and what is the cost of reflections?
6. Safety concerns: do GPT models always fulfill users' requests, and is there any case that the model refuses to fulfill the request or gives out a warning due to safety concerns of the generation process or the chemical itself, or is there any case that the model fails to generate the answer due to its limited knowledge of certain uncommon chemicals?

---

> ### Author Response · Authors · 2024-11-20
>
> ## (1/3)
>
> We thank you for recognizing the **originality** and **effectiveness** of our proposed OSDA Agent, as well as the **novelty** of the self-reflection mechanism in optimizing molecule generation.  We are grateful for your constructive comments, which have contributed to improving the clarity and impact of our work.
>
> Below, we provide detailed answers to the reviewer's concerns.
>
> # Weaknesses1： Chemical Safety Assessment
>
> Thank you for your valuable comments. We completely agree with the importance of safety in chemical molecule design. Since safety assessments require specialized chemical tools, we plan to incorporate chemical safety evaluation tools such as Toxtree[1] and EXPLO5[]2] to assess the molecules we design. These tools are effective in identifying potential safety concerns, such as toxicity and explosiveness, and issuing warnings for molecules that may present risks. By integrating these tools, we aim to further refine our approach, ensuring that the generated molecules not only meet the chemical property requirements but also adhere to chemical safety standards.
>
> [1]Toxtree is a software tool used for predicting the toxicity of chemical substances through structure-toxicity relationship (SAR) models.
>
> [2]EXPLO5 is a software tool used for calculating the thermodynamic properties and phase behavior of chemical substances, and it can be used to assess the explosive properties of molecules.
>
> # Question1：Support for Multiple Inputs
> Currently, our model primarily supports converting chemical input into SMILES (Simplified Molecular Input Line Entry System) strings, as SMILES is a standardized and widely used molecular representation. This format is also more compatible with the chemical tools employed in our evaluator, such as the energy estimation models， Void and Scscore, which both take SMILES as input.
>
> Regarding IUPAC names, while our current implementation does not directly handle them, we recognize the value of supporting multiple chemical expressions. We can consider integrating additional tools or leveraging the capabilities of the large language model itself to convert IUPAC names into SMILES strings, enabling flexibility in the input format.
>
> In summary, while the evaluator currently works with SMILES input, we are open to exploring ways to expand the model's ability to handle other chemical representations in the future.
>
> # Question2：Accuracy of binding energy estimation
>
> Thank you for your thoughtful question. To evaluate the accuracy of our binding energy estimation model, we trained and tested it using the OSDB database, where the binding energy values are derived from traditional atomic simulation methods. These values serve as our ground truth labels.
>
> Our model achieved a Mean Absolute Error (MAE) of approximately 0.38 kcal/mol, which demonstrates a high level of accuracy. This performance is comparable to existing binding energy estimations in the OSDA framework. For example, for AFI-type zeolites, the binding energy ranges from -3 kcal/mol to -9 kcal/mol, further demonstrating the model's robustness, particularly for this class of materials.
>
> We believe this level of accuracy provides strong evidence that our model can be a reliable alternative to more computationally expensive traditional atomic simulation methods, especially in cases where reducing computational complexity is critical.

---

> ### Author Response · Authors · 2024-11-20
>
> ## (2/3)
>
> # Question3：Explore Open Source LLM
>
> We appreciate your suggestion to explore open-source models such as Llama[1] and Mistral[2]. We did indeed experiment with both models and compared their performance against OpenAI's GPT-4 in the context of our OSDA (Open Source Design Agent) for chemical design tasks.
>
> | Method                   | BLEU $\uparrow$ | Morgan $\uparrow$ | MACCS $\uparrow$ | RDK $\uparrow$ | ED $\downarrow$ | KL Divergence $\downarrow$ |
> |--------------------------|-----------------|-------------------|------------------|----------------|-----------------|----------------------------|
> | OSDA Agent               | 0.601          | 0.368            | 0.816           | 0.624         | 0.934          | 0.825                     |
> | Llama                    | 0.522          | 0.301            | 0.628          | 0.416         | 1.693           | 1.073                      |
> | OSDA Agent (Llama)       | 0.551    $\uparrow$      | 0.315     $\uparrow$       | 0.755     $\uparrow$      | 0.565     $\uparrow$    | 0.901  $\uparrow$        | 0.693        $\uparrow$             |
> | Mistral                  | 0.338          | 0.177            | 0.411           | 0.224         | 2.692           | 1.192                      |
> | OSDA Agent (Mistral)     | 0.512       $\uparrow$   | 0.306   $\uparrow$        | 0.740    $\uparrow$       | 0.541 $\uparrow$        | 0.825   $\uparrow$       | 0.661       $\uparrow$              |
>
>
> [1]https://huggingface.co/mistralai/Mistral-Nemo-Instruct-2407
>
> [2]https://huggingface.co/meta-llama/Llama-3.1-70B-Instruct
>
> The OSDA Agent is our default model. The OSDA Agent (Llama) is fully built on Llama, OSDA Agent (Mistral) is fully built on Mistral.
> Our experimental results show that, regardless of the model used, our OSDA Agent significantly enhances the overall design results (**OSDA Agent (Llama)** Vs **Llama**, **OSDA Agent (Mistral)** Vs **Mistral**). However, currently, its performance is still slightly inferior to GPT-4, which remains the most effective model for this specific task.
>
> # Question4：About the evaluation index "Validity"
>
> The validity metric evaluates the proportion of generated molecules that meet a set of screening criteria derived from decades of research on OSDAs. These criteria, developed by domain experts, include empirical rules regarding molecular rings, bond types, functional groups, and elemental compositions. Chemists typically avoid attempting to use molecules that do not satisfy these rules in zeolite synthesis, making this metric highly relevant for practical applications.
>
> Regarding performance, our approach uses chemical tools to actively screen generated molecules during the generation process. Molecules that do not meet the criteria are flagged, and the model reflects on and regenerates them, ensuring that all output molecules adhere to these screening rules. This process contributes to the high validity score achieved by our model.
>
> In contrast, baseline text-based molecular design methods, such as BIOT5 and MOLT5, were tested using textual inputs tailored to specific zeolites and screening criteria translated into natural language. However, due to the lack of specialized domain knowledge and limited understanding of nuanced natural language descriptions, these baseline models rarely generated molecules that met the validity criteria.

---

> ### Author Response · Authors · 2024-11-20
>
> ## (3/3)
> # Question5：About the Reflection  Count
>
> **Would more reflections yield better performance?**
> Based on our observations, during the molecule optimization task, the first few reflections (typically the first four to five) effectively improve the SCScore and binding energy. This is achieved by optimizing the carbon chain length, functional group positioning, simplifying the molecular backbone, and reducing steric hindrance, while maintaining the functional characteristics of the original molecule. However, starting from the fifth to the sixth iteration, the optimized molecules begin to significantly differ in structure from the original molecule, and the estimated binding energy starts to rise. This suggests that while more reflections may improve the binding energy further, they may sacrifice functional consistency with the original molecule. Therefore, we set the number of reflections to 4-5 in most of our experiments to balance performance enhancement and functional consistency.
>
> **What is the cost of reflections?**
> The primary cost of each reflection is associated with the evaluation of the generated molecules. Specifically, each reflection typically incurs a time cost of 1 to 3 minutes. In our experimental setup, each reflection involves modifying the original molecule to optimize the SCScore and binding energy, which requires evaluating the performance of the new molecule. As a result, increasing the number of reflections leads to higher computational costs, especially when multiple iterations are involved.
>
> In summary, while increasing the number of reflections might improve the binding energy further, too many reflections may reduce functional consistency with the original molecule. Additionally, each reflection incurs a computational cost. In our experiments, we found that setting the number of reflections to 4-5 strikes a good balance between performance improvement and computational cost.
>
> # Question6： About the security of LLM
>
> To date, we have not encountered any issues where the LLM fails to respond due to safety concerns related to the generated chemical molecules. However, we acknowledge that assessing the safety of chemical molecules—such as toxicity and explosiveness—requires more specialized chemical tools, like Toxtree and EXPLO5. It is difficult for LLMs to determine the toxicity or explosiveness of a molecule solely based on its molecular structure. For instance, the molecule we generated, C\[N+\]1(C)CC2CCC1CC2, was identified as toxic by the Toxtree tool, but the LLM did not issue a warning during the generation process. We plan to incorporate additional chemical tools in the future to assess and provide warnings regarding the safety of the generated chemical molecules.
>
> Currently, we have not encountered a situation where the LLM completely refuses to generate content. While the generated molecules may not meet the requirements in various aspects, we utilize a reflection process to allow the LLM to reconsider and generate molecules that better align with the desired specifications.

---

> > ### Comment · Reviewer_t8Dz · 2024-11-25
> > **Thank you for addressing my questions**
> >
> > Thank you so much for your detailed responses to my questions and concerns. I have a better understanding of the paper and decided to adjust my score.

---

> > > ### Author Response · Authors · 2024-11-25
> > > **Response to Reviewer t8Dz**
> > >
> > > Dear Reviewer t8Dz,
> > >
> > > We appreciate that our rebuttal addressed your concerns. Also, thank you for the support for our work! Please let us know if you have any further questions..

---

### Official Review · Reviewer_bUa4 · 2024-11-04

**Soundness:** 3
**Presentation:** 3
**Contribution:** 3
**Rating:** 8
**Confidence:** 3

**Summary:**

This paper introduces OSDA Agent, a novel framework that combines LLMs (particularly GPT-4) with computational chemistry tools to design Organic Structure Directing Agents (OSDAs) for zeolite synthesis. The framework consists of three key components:

1. An Actor powered by GPT-4 that designs potential OSDAs using few-shot Chain of Thought prompting, In-Context Learning with OSDB database examples, and a memory component for context.

2. An Evaluator that employs computational chemistry tools (RDKit for validity, SCScore for synthesis feasibility) and a novel binding energy model(having an MAE of 0.384 kJ/mol Si) to assess generated structures.

3. A Self-Reflector using GPT-4o that provides iterative feedback based on evaluations stored in memory to improve chemical validity, synthetic feasibility, and binding energy estimations through a generation-evaluation-reflection-refinement workflow.

The authors validate their framework across multiple zeolite types, demonstrating its ability to generate chemically valid OSDA candidates outperforming the baselines across multiple similatiy metrics. Additionally, the framework successfully optimizes existing OSDA molecules, reducing their synthetic complexity scores from 3.45 to 2.46 while maintaining their functional properties and keeping binding energies within desired ranges (-3.38 to -9.00 kcal/mol). The authors benchmark their experiments against other methods such as MolT5 and BioT5 using diverse metrics including validity, MACCS, and BLEU scores. Subject matter experts have confirmed that the OSDAs designed by the OSDA Agent show the potential to function effectively as structure-directing agents.

**Strengths:**

- Novel OSDA framework that combines LLMs with chemical validation tools and binding energy models.
- Immense practical significance due to zeolites' widespread industrial applications, making efficient OSDA generation methods valuable.
- Strong experimental results demonstrating OSDA Agent's ability to generate competitive and sometimes superior OSDA designs over baseline methods such as MolT5, BioT5, and pure GPT-4
- Comprehensive evaluation using multiple metrics (validity, similarity, distribution measures) strengthened by validation from domain experts.
- Demonstrated capability to optimize existing OSDA structures by reducing synthetic complexity scores while maintaining functional properties.

**Weaknesses:**

- Additional experiments examining the relative importance of different components (e.g., the reflector mechanism) and various tools (RDKit, SCScore, binding energy models), along with analysis of failure patterns (such as consistent failures in chemical validity) would strengthen the paper's findings.
- The paper does not specify the number of iterations used in the design-evaluate-reflect process.  Additional analysis of how metrics change with iterations and when the performance plateaus would be valuable.
- A comparison with domain-specific fine-tuned LLMs or using LLMs other than GPT-4 could strengthen the generalizability of the work.

**Questions:**

- Could you specify the number of iterations of the OSDA agent (design-evaluate-reflect iterations)? What criteria did you use to determine the optimal number of iterations?

- The paper demonstrates OSDA Agent's success on multiple zeolite types, how does your framework handle increasing zeolite complexity? Are there any known limitations in terms of zeolite structure complexity or OSDA size?

- Could you elaborate on the expert validation process? Specifically, what validation criteria were used, and have any of your generated OSDAs been experimentally synthesized?

- Your framework integrates multiple components (reflection mechanism, multiple aspects in the evaluator such as RDKit, SCScore, and binding energy models). Have you conducted ablation studies to understand their relative importance? For instance, how much does the reflection mechanism contribute to the final performance, and what are the most common types of failures?

- What motivated the choice of using different LLM variants (GPT-4 for the Actor, GPT-4o for the Self-reflector) in your framework? Have you conducted comparative experiments with other LLM combinations, either using the same model for both components or testing with open-source LLMs?

---

> ### Author Response · Authors · 2024-11-20
>
> ## (1/3)
>
> We would like to express our sincere gratitude for your recognition of the **originality** and **practical significance** of our work, as well as for your acknowledgment of the **effectiveness** of the OSDA Agent framework we proposed. We also appreciate your constructive suggestions, which will be invaluable in helping us further improve and refine our approach.
>
> Below, we provide detailed answers to the reviewer's concerns.
>
> # Part I:  Ablation Study on Model Components
> （Weaknesses1 and question4）
>
> In this paper, the core of the proposed OSDA Agent method lies in the **reflection mechanisms**, where the chemical tools provide reflective information from different aspects. The results of the ablation study are presented below：
>
> | Method                     | Validity $\uparrow$ | BLEU $\uparrow$ | Morgan $\uparrow$ | MACCS $\uparrow$ | RDK $\uparrow$ | ED $\downarrow$ | KL Divergence $\downarrow$ | Avg Rank |
> |----------------------------|---------------------|-----------------|-------------------|------------------|----------------|-----------------|----------------------------|----------|
> | OSDA Agent                 | **1.000**              | 0.601          | 0.368            | **0.816**           | **0.624**         | **0.934**          | **0.825**                    | **1.28**     |
> | Remove reflection mechanism | 0.702              | 0.581          | 0.331            | 0.782           | 0.553         | 1.359          | 0.973                    | 4.57     |
> | Remove RDKit                | 0.770              | 0.593         | 0.355            | 0.751           | 0.566         | 1.233          | 0.830                   | 3.42     |
> | Remove Scscore              | 1.000              | 0.570         | **0.372**            | 0.802          | 0.614         | 1.256          | 1.001                     | 2.85     |
> | Remove blending energy      | 1.000             | **0.627**          | 0.356            | 0.787           | 0.619         | 1.275          | 0.972                     | 2.42     |
>
>
>
> Based on the results of the ablation study, when measuring importance by average ranking, the component with the most significant impact on our method is the **reflection mechanisms**. For our approach, if removing the reflection mechanisms, our model will degrade into using In-Context learning and few-shot Cot prompt engineering for OSDA molecular design. Removing this part leads to the poorest average performance.  The next most impactful component is the **RDKit tool**. Furthermore, the **SCScore**(Synthetic Complexity Score) and **Binding Energy** models have a considerable influence on the molecular geometry (It has a great influence on WHIM energy distance and KL divergence)
>
> The most common type of failure in our experiments is the **deviation from the required C/N ratio**. We speculate that the reason for this is that the C/N ratio is a clear and quantitative constraint in molecular design, but the LLM lacks direct numerical calculation capabilities, leading to these failures.
>
> # Part II: Reflection Iteration Count Setting
> （Weaknesses2 and question1）
>
> Thank you for your insightful comment. When choosing the number of iterations, we considered experimental outcomes, stability, and computational resource consumption. Our experiments consisted of two task: molecular optimization and molecular design, with the iteration count based on the following analysis.
>
> In the molecular optimization task, we fixed the initial molecule and optimized its SCscore(Synthetic Complexity Score) and Binding Energy.  In the first 4 iterations, optimization showed significant improvement, and the molecular functionality remained stable. However, from the 5th iteration onwards, the optimized molecule showed considerable structural differences from the initial molecule, indicating potential over-optimization that could deviate from the intended goal. Therefore, we limited the iterations to 4 to avoid further structural changes.
>
> In the molecular design task, where no initial molecule was fixed, we relied on the OSDA agent to design new molecules. After 4-5 iterations, SCscore stabilized between 1.7 and 1.8, and Binding Energy reached a stable level. After the 6th iteration, SCscore began to fluctuate, and Binding Energy increased, suggesting diminishing returns from further iterations. Thus, setting the iterations to 4-5 ensures stable results while avoiding unnecessary fluctuations or performance decline.
>
> Our OSDA molecular design framework is an automated chemical process, where time and cost are critical. Each iteration adds computational expense, with excessive iterations yielding diminishing returns. Thus, 4-5 iterations strike a balance between optimization and efficiency.
>
> In conclusion, the choice of 4-5 iterations is based on our experimental observations, balancing stable results with computational cost, ensuring both efficiency and sustainability.

---

> ### Author Response · Authors · 2024-11-20
>
> ## (2/3)
>
> # Part III: Comparison of Components Across Different Large Language Models
>
> （Weaknesses3 and question 5）
>
>  The decision to use different LLM variants (GPT-4 for the Actor and GPT-4o for the Self-reflector) was motivated by the need to leverage the extensive chemical knowledge embedded in large models, especially in the design phase of OSDA. The Actor is responsible for summarizing and synthesizing information for molecular design, and it benefits from GPT-4's advanced capabilities in this area. In addition, the reflection process plays a crucial role in refining the molecule designs, and GPT-4’s performance in this phase is highly effective. To balance computational efficiency, we used GPT-4o as the Self-reflector, which offers a cost-effective solution while maintaining the required functionality for reflecting and improving the designs.
>
> We acknowledge the importance of testing with Open Source LLMs to enhance the generalizability of our framework. We have also demonstrated that our approach can significantly improve the OSDA design capabilities using open-source models. Specifically, we tested two popular open-source LLMs, Mistral[1] and LLama 3.1[2], and found that our method also yielded substantial improvements in their ability to design OSDA molecules
>
> | Method                   | BLEU $\uparrow$ | Morgan $\uparrow$ | MACCS $\uparrow$ | RDK $\uparrow$ | ED $\downarrow$ | KL Divergence $\downarrow$ |
> |--------------------------|-----------------|-------------------|------------------|----------------|-----------------|----------------------------|
> | OSDA Agent               | 0.601          | 0.368            | 0.816           | 0.624         | 0.934          | 0.825                     |
> | OSDA Agent*              | 0.571          | 0.317            | 0.772           | 0.601         | 0.964          | 1.091                      |
> |--------------------------|-----------------|-------------------|------------------|----------------|-----------------|----------------------------|
> | Llama                    | 0.522          | 0.301            | 0.628           | 0.416         | 1.693          | 1.073                      |
> | OSDA Agent (Llama)       | 0.551 $\uparrow$         | 0.315 $\uparrow$           | 0.754   $\uparrow$        | 0.565 $\uparrow$        | 0.901  $\uparrow$        | 0.693     $\uparrow$                |
> | Mistral                  | 0.338          | 0.177            | 0.411           | 0.224         | 2.692           | 1.192                      |
> | OSDA Agent (Mistral)     | 0.512    $\uparrow$      | 0.306  $\uparrow$          | 0.740  $\uparrow$         | 0.541 $\uparrow$        | 0.825   $\uparrow$       | 0.661   $\uparrow$                  |
>
>
> The OSDA Agent is our default model. The OSDA Agent* replaces the actor with GPT-4o while the OSDA Agent (Llama) is fully built on Llama, OSDA Agent (Mistral) is fully built on Mistral.
>
> Our experimental results show that, regardless of the model used, our OSDA Agent significantly enhances the overall design results(**OSDA Agent (Llama)** Vs **Llama** , **OSDA Agent (Mistral)** vs **Mistral** ). However, currently, its performance is still slightly inferior to GPT-4, which remains the most effective model for this specific task.
>
> [1]https://huggingface.co/mistralai/Mistral-Nemo-Instruct-2407
>
> [2]https://huggingface.co/meta-llama/Llama-3.1-70B-Instruct
>
> # Part IV: Addressing the Complexity of Zeolite Structures and OSDA Size Limitations
>
> （question2）
>
> As of now, there are approximately 250 types of zeolites, and our dataset includes 210 zeolite structures, covering the vast majority of zeolite frameworks. The seven zeolites designed in this paper are also representative types used in industry. Within different zeolite structures, based on chemical field experience, the pore structure, pore size, thermal stability, and framework type have a significant impact on OSDA selection. For new zeolite structures, we can use examples of OSDA from similar zeolite frameworks to design new prompts based on the characteristics of the new structure. When training the binding energy model, we can consider purposeful data augmentation and perturbation to improve generalization ability.
>
> Zeolite OSDAs generally refer to small molecules, and our OSDA agent is currently focused on small molecules. For large molecules (with a molecular weight greater than 1000 ), such as surfactants, their interaction with zeolites differs significantly from small molecules. Therefore, Therefore, the screening criteria in Table 2 of the paper and the binding energy estimates will no longer be valid for such large molecules.

---

> ### Author Response · Authors · 2024-11-20
>
> ## (3/3)
>
> # Part V: Expert Validation
>
> （question 3）
>
> We submitted the generated molecules to several experts in the field of chemistry, who used their extensive knowledge and practical synthesis experience to evaluate the submitted molecules. The focus was on the rationality of the newly designed OSDA molecules and the feasibility of their synthesis.
>
> Below are several examples of expert evaluation
>
> generated：CC[N@+]1(C)CCCCC1
>
> The generated compound and the literature compound CCCC[N+]1(C)CCCC1 [1] are both cyclic quaternary ammonium salts. The ring structures are six-membered rings and five-membered rings. These two ring structures have similar tension and stability in terms of organic chemistry, and both are stable. The N atom in both molecules contains methyl and flexible low-carbon alkyl chains. The generated one has fewer carbon atoms in the chain and is easier to synthesize.
>
> generated：CC\[N+\](C)(C)C(C)C
>
> The generated compound and the literature compound CCC\[N+\](C)(C)CCC [2] both are chain quaternary ammonium salts. They also contain the same number of carbon atoms. The structure difference is that one contains an n-propyl group and the other contains an isopropyl group. Yet n-propyl group and the isopropyl group have similar properties in organic compounds. The generated is easy to synthesize or purchase commercially.
>
> generated：C\[N+\]1(C)C[C@H]2CC\[C@H\](CC2)C1
> The generated and the literature compound CC1(C)CC2CC(C)(C1)C\[N+\]2(C)C[3] both bridged ring quaternary ammonium salts. The bridged ring structures in the two molecules have similar tensions and stabilities in terms of organic chemistry. The N atom in both molecules contains the methyl group, and thus has similar chemical properties. The literature molecule can be transformed into the generated one by demethylation and ring cleavage reactions.
>
> The experimental synthesis of these molecules requires a certain amount of time, and we are currently unable to provide specific synthesized examples. However, given that the generated molecules have been validated and endorsed by domain experts, along with their low synthesis complexity scores, we believe there is a high likelihood of successful synthesis for these molecules.
>
>
> [1]:Azim M M, Stark A. Ionothermal synthesis and characterisation of Mn-, Co-, Fe-and Ni-containing aluminophosphates[J]. Microporous and Mesoporous Materials, 2018, 272: 251-259.
>
> [2] Lee J H, Kim Y J, Ryu T, et al. Synthesis of zeolite UZM-35 and catalytic properties of copper-exchanged UZM-35 for ammonia selective catalytic reduction[J]. Applied Catalysis B: Environmental, 2017, 200: 428-438.
>
> [3]Wagner P, Nakagawa Y, Lee G S, et al. Guest/host relationships in the synthesis of the novel cage-based zeolites SSZ-35, SSZ-36, and SSZ-39[J]. Journal of the American Chemical Society, 2000, 122(2): 263-273.

---

> > ### Comment · Reviewer_bUa4 · 2024-11-27
> >
> > Thank you for the detailed explanations of all my queries, questions and concerns!

---

> > > ### Author Response · Authors · 2024-11-27
> > > **Response to Reviewer bUa4**
> > >
> > > Dear Reviewer bUa4,
> > >
> > > We appreciate that our rebuttal addressed your concerns. Please let us know if you have any further questions.

---

### Official Review · Reviewer_93jd · 2024-11-04

**Soundness:** 3
**Presentation:** 3
**Contribution:** 3
**Rating:** 8
**Confidence:** 4

**Summary:**

The paper introduces the OSDA Agent, an innovative framework combining Large Language Models (LLMs) with computational chemistry tools for the de novo design of Organic Structure Directing Agents (OSDAs) for zeolites. The framework includes three main components: Actor, Evaluator, and Self-reflector, enhancing the process of generating and optimizing OSDA molecules. Demonstrates significant improvements over existing models in generating OSDAs that are not only theoretically valid but practically feasible. Provides a comprehensive methodological approach and substantial experimental results, setting a new standard in computational chemistry and molecular design.

**Strengths:**

1. Integrating LLMs with computational chemistry tools to create a feedback-informed molecule generation model is unique. This model addresses the limitations of traditional molecule design models by providing an interactive and iterative design process that includes a novel binding-energy prediction module (Section 4.1).
2. The experimental design is robust, with clear definitions of metrics and methodological approaches that ensure the reproducibility and validity of results. The use of a layered approach combining different computational tools to assess molecule viability is particularly noteworthy (Sections 5.1 and 5.2).
3. The paper is well-structured and written, offering detailed explanations of the processes and technologies involved, such as the Actor-Evaluator-Self-reflector framework. Figures and tables effectively illustrate complex processes and results.
4. This research has a high potential impact, particularly on industries relying on zeolites. Generating and optimizing OSDAs more efficiently could lead to significant advancements in material science and related fields.

**Weaknesses:**

1. The paper could better address how the proposed model might generalize to other types of molecular design beyond OSDAs for zeolites. It is unclear whether the techniques and improvements reported apply to broader classes of materials or chemical processes.
2. The reliance on external computational chemistry tools may introduce limitations related to the scalability and speed of the proposed approach. The paper could expand on the implications of these dependencies, particularly in terms of computational cost and accessibility.
3. The models are trained and validated primarily using the Zeolite Organic Structure Directing Agent Database and the Jensen dataset. The diversity and representativeness of these datasets could be questioned, potentially affecting the robustness and applicability of the findings (Section 3.1).

**Questions:**

1. Can the methodologies and framework presented be adapted to design other types of molecules or materials not discussed in the paper? What changes or adaptations would be necessary?
2. What are the computational resource requirements for implementing the OSDA Agent framework, significantly when scaling to larger datasets or more complex molecular structures?
3. How does the framework handle changes or updates in computational chemistry tools or LLMs? Is there a mechanism to easily integrate new tools or updates in knowledge of the field?
4. Could you elaborate on the choice of evaluation metrics used? How do these metrics align with the practical requirements of OSDA applications in the industry?

---

> ### Author Response · Authors · 2024-11-20
>
> ## (1/3)
>
> We thank you for your kind acknowledgment of our work as **unique** and **innovative**, and for recognizing the **novelty** of our approach in integrating LLMs with computational chemistry tools. We thank you for your constructive comments.
>
> Below, we provide detailed answers to the reviewer's concerns.
>
> # Generalizability of the Proposed Method
>
> （Weaknesses 1）
>
> Thank you for your valuable feedback. We agree that the generalizability of our proposed model is an important consideration. While the current study focuses specifically on the design of OSDAs for zeolites, we believe that the underlying methodology can be extended to other types of molecular design. The model is built upon a foundation of large language models (LLMs) trained on a diverse range of scientific texts, providing it with **fundamental chemical knowledge** that supports broader molecular design capabilities. Additionally, the LLM Agent is highly **flexible and adaptable**, capable of selecting appropriate tools based on specific design requirements. This adaptability enables the model to be applied to broader classes of materials and chemical processes, beyond zeolites and OSDAs.
>
> # Computational cost and Accessibility
>
> （Weaknesses 2）
>
>  The computational chemistry tools we rely on—RDKit, Scscore, and Void—are all accessible via Python, which allows for seamless integration within our framework. Furthermore, the computational chemistry component of our model is modular, meaning we have the flexibility to choose or swap tools as needed without requiring changes to the overall structure of the framework. This modularity ensures that the approach can adapt to future updates or replacements of the computational tools, offering flexibility and scalability.
>
> In terms of computational cost, our method primarily incurs time costs related to evaluating the generated molecules, particularly when Void uses Voronoi diagrams to sample docking poses. Each OSDA-zeolite pair can generate tens to hundreds of docking configurations. Our experiments show that the evaluation time for each molecule is approximately 1 to 3 minutes.
>
> # Diversity and Representativeness of the Data
>
> （Weaknesses 3）
>
> We understand the concern about the diversity and representativeness of the datasets used in training and validation, and we appreciate the opportunity to clarify this point. We selected the Zeolite Organic Structure Directing Agent Database (OSDA Database) and the Jensen dataset primarily because they are both widely recognized and extensively applied in the field of zeolite materials and organic structure-directing agents (OSDAs). The Jensen dataset, which compiles 5,663 zeolite synthesis pathways extracted from 1,384 publications between 1966 and 2020, is one of the most comprehensive datasets in this domain. It covers a broad range of zeolite types and provides a wide variety of OSDA molecular volumes,  ensuring a certain level of diversity. Similarly, the OSDB (OSDA Database) is built upon first-principles simulations, providing composite energy data for over 500,000 zeolite-OSDA pairs. To the best of our knowledge, this is one of the most complete datasets of its kind in terms of binding energy data. In this paper,the types of zeolites tested in our experiments are also representative of various zeolite types used in industry.
>
> While we acknowledge that no dataset is exhaustive, we believe that these datasets are sufficiently representative of the range of zeolite types and OSDAs commonly encountered in the field, which supports the robustness and applicability of our findings. The use of these well-established datasets also ensures that our model is grounded in widely accepted data within the community.

---

> ### Author Response · Authors · 2024-11-20
>
> ## (2/3)
>
> # Adapting the Methodologies and Framework for Designing Other Molecules or Materials
>
> （Question1）
>
> We believe that the methodologies and framework presented in the paper can indeed be adapted to design other types of molecules or materials, though some adjustments would be necessary to account for the specific requirements of different domains.
>
> **Prompt Engineering**, previous studies have shown that providing large language models (LLMs) with context through In-Context Learning (ICL) and guiding the reasoning process using methods like Chain-of-Thought (CoT) can enhance their ability to solve complex problems. For designing other types of molecules or materials, it would be necessary to carefully design the prompts according to the specific problem at hand. This may require domain-specific expertise to ensure that the model is given the correct context and instructions for the new design task. Fine-tuning the prompts will be essential for effectively applying the model to different molecular or material design challenges.
>
> **Selecting Task-Specific Tools**， as LLMs can sometimes generate outputs that are factually incorrect, commonly referred to as "hallucinations," it is crucial to integrate specialized chemical tools to ensure the generated results are accurate and meet the desired criteria. By incorporating targeted computational tools and databases that are tailored to the new materials or molecules being designed, we can validate the outputs and improve the reliability of the model’s predictions. These tools will help filter out erroneous results and ensure that the design process is aligned with the specific requirements of the new system.
>  While the core methodology is adaptable to other molecular or material design tasks, appropriate adjustments in prompt design and the integration of specialized chemical tools will be necessary to ensure the framework can meet the specific demands of different applications.
>
> # Computational Resource Requirements and Scaling Molecule Type
>
> （Question2）
>
> The OSDA Agent framework is designed to efficiently handle typical molecular design tasks. The chemical tools we use, namely Void and Scscore, are both Python-based and primarily rely on CPU computation. However, to improve efficiency, particularly when estimating binding energies, we leverage A6000 GPU for more intensive computations.
>
> For tasks that involve generating a large number of OSDA candidate molecules, we can control the number of docking poses generated by Void, which helps manage computational load and reduce the time spent on energy estimations.
>
> Regarding more complex molecular structures, our observations suggest that the complexity of OSDA molecular structures has a minimal impact on the runtime of Scscore and Void. These tools perform relatively consistently across different levels of molecular complexity, making the framework scalable even for more intricate molecular designs.
>
> # Handling Updates and Integration of New Tools in the Framework
>
> （Question3）
>
> We recognize that the fields of computational chemistry and machine learning are evolving rapidly, and our OSDA Agent framework is designed with flexibility and extensibility in mind to accommodate this change. The LLM Agent, as well as the computational chemistry components, are built with a modular architecture that allows for easy integration of new tools and updates.
>
> **Modular Design for Computational Tools**: The computational chemistry portion of the framework is a standalone module. This design allows us to update or replace any specific computational tool (e.g., Void, Scscore) by simply modifying the interfaces, without requiring significant changes to the overall framework. This modular approach ensures that updates or substitutions of tools can be seamlessly incorporated, preserving the integrity of the entire system.
>
> **Updating Knowledge and LLMs**: In addition to accommodating updates in computational tools, we also ensure that the framework stays up-to-date with the latest developments in chemical knowledge. We achieve this by continuously expanding and updating the relevant databases. For example, new data can be added to the training sets and incorporated into the prompt engineering process to improve the model's predictive capabilities. As LLMs evolve and improve, we can integrate new versions or fine-tuned models to further enhance the system's performance.
>
> In summary, the OSDA Agent framework is designed to be highly adaptable, with a modular structure that facilitates the easy integration of new computational chemistry tools and LLMs. This approach ensures that the framework can keep pace with technological advancements and maintain its relevance in the face of ongoing developments in both computational chemistry and machine learning.

---

> ### Author Response · Authors · 2024-11-20
>
> ## (3/3)
>
> # Evaluation Metrics and Their Relevance to OSDA Applications
>
> （Question4）
>
> The metrics we selected are specifically designed to address the key practical requirements for OSDA applications in industry, and are divided into three main categories: **validity**, **molecular similarity**, and **distribution similarity**. Additionally, we incorporate subjective expert judgment to further ensure the relevance of the generated molecules in real-world OSDA design.
>
> **Validity**: In decades of OSDA research, chemists have identified several empirical rules for selecting potential OSDA molecules, focusing on characteristics such as ring structures, bond types, functional groups, and elemental compositions. Molecules that do not meet these criteria are generally not considered viable candidates for OSDA design. Our Validity metric is aimed at quantifying how well the generated molecules satisfy these established criteria. It represents the proportion of generated molecules that conform to the structural and chemical requirements necessary for OSDA molecules, making it highly relevant for practical applications where these rules are well-established.
>
> **Molecular Similarity**: We use several well-established molecular similarity metrics, including BLEU, Morgan, MACCS, and RDK, which are commonly used in molecular design tasks. These metrics assess the similarity between the generated molecules and existing molecules, focusing on different aspects of the chemical structure. In traditional OSDA design, chemists often look for molecules with similar structures or functional groups to known OSDAs, so these similarity metrics align with how OSDA molecules are typically identified in practice.
>
> **Distribution Similarity**: For distribution similarity, we introduced two novel metrics: Molecular WHIM energy distance (ED) and Kullback-Leibler divergence (KL). These metrics are designed to evaluate the similarity in distribution between the generated molecules and existing OSDAs, particularly in terms of their geometric and topological properties. The WHIM descriptor focuses on a molecule’s geometric features, such as size, atomic distances, angles, and topological structure. Since specific zeolite pores have fixed sizes, the molecules that can bind to these pores must have similar geometric properties. Therefore, the WHIM energy distance and KL divergence metrics help us assess how well the generated molecules match the structural distributions of known OSDAs, which is crucial for real-world applications where the geometric compatibility of OSDAs with zeolite pores is critical.
>
> **Expert Judgement**: Finally, given that OSDA design still relies on some degree of experience and intuition, we also incorporated expert judgment to subjectively evaluate the generated molecules. This step ensures that our generated candidates align with practical and empirical insights from the field of OSDA design.

---

### Author Response · Authors · 2024-11-20
**Summary of Revision**

Dear Chairs and Reviewers,

We would like to thank the reviewers for their careful and constructive comments. We also thank the reviewers for acknowledging our work is original, effective ( 93jd,t8Dz,WsXS
), novel ( bUa4,93jd) and has great potential (93jd). The paper has been revised in accordance with the reviewers’ comments and suggestions. Updates and changes are marked by blue color in the revised version. The major changes in this revision lie in the following aspects:

Appendix B:Ablation Study

Appendix G:Evaluation of Alternative LLM Components

Appendix H:Supplementing the Background and Motivation

Appendix J:Definition and Explanation of Terms and Concepts in the Paper.

Should you need further information, please let us know. We look forward to hearing from you soon.

Yours sincerely,

Authors of Paper “OSDA Agent: Leveraging Large Language Models for De Novo Design of Organic Structure Directing Agents”

---

### Meta-Review · Area_Chair_SLeM · 2024-12-18

**Metareview:**

This paper presents an LLM-driven framework (combined with computational tools) for designing OSDA molecules for zeolite synthesis, integrating computational chemistry and a reflection component to refine candidates iteratively. The reviewers find the approach original, effective, and of practical significance. Reviewers noted the experimental results are strong with clear improvements over baseline methods. Some reviewers requested clearer explanation of why OSDAs are challenging to design and how certain components contribute, the authors provided detailed clarifications, ablation results, and additional background in their rebuttal.

Overall, the paper’s contributions—combining LLMs, chemistry tools, and an iterative refinement cycle—represent a meaningful step forward for molecular design tasks. The paper addresses reviewer concerns adequately, and I recommend acceptance. There are additional prior work on applying LLMs, computational tools, machine learning potentials, DFT calculations, etc in a pipelined/iterative approach for iterative search/refinement of other types of materials [1,2,3]. I recommend the authors consider including these work in their discussion.

[1] FlowLLM: Flow Matching for Material Generation with Large Language Models as Base Distributions
[2] Fine-Tuned Language Models Generate Stable Inorganic Materials as Text
[3] Generative Hierarchical Materials Search
[4] MatterGen: a generative model for inorganic materials design

**Additional Comments On Reviewer Discussion:**

The reviewers raised questions around motivation and clarity, generality and novelty of the approach, and insight into dataset limitations and choices of baseline models. The authors clarified the complexity and limited availability of OSDA examples, explained the significance of feedback-driven iterative design, and provided ablation studies to highlight the importance of each component. The discussions, revisions, clarifications sufficiently resolved major concerns.

---

### Decision · Program_Chairs · 2025-01-22

Accept (Spotlight)